# CoVAE: Consistency Training of Variational Autoencoders

## Abstract

Current state-of-the-art generative approaches frequently rely on a two-stage training procedure, where an autoencoder (often a VAE) first performs dimensionality reduction, followed by training a generative model on the learned latent space. While effective, this introduces computational overhead and increased sampling times. We challenge this paradigm by proposing Consistency Training of Variational AutoEncoders (CoVAE), a novel single-stage generative autoencoding framework that adopts techniques from consistency models to train a VAE architecture. The CoVAE encoder learns a progressive series of latent representations with increasing encoding noise levels, mirroring the forward processes of diffusion and flow matching models. This sequence of representations is regulated by a time dependent $\beta$ parameter that scales the KL loss. The decoder is trained using a consistency loss with variational regularization, which reduces to a conventional VAE loss at the earliest latent time. We show that CoVAE can generate high-quality samples in one or few steps without the use of a learned prior, significantly outperforming equivalent VAEs and other single-stage VAEs methods. Our approach provides a unified framework for autoencoding and diffusion-style generative modeling and provides a viable route for one-step generative high-performance autoencoding. Our code is available in the supplementary material.

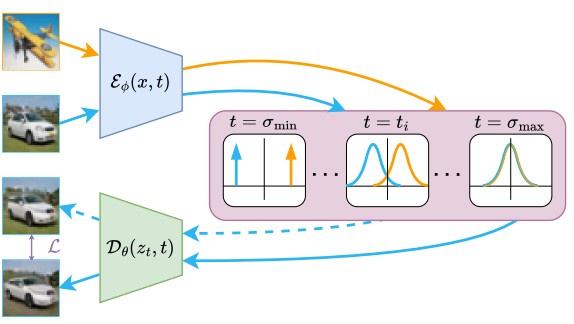 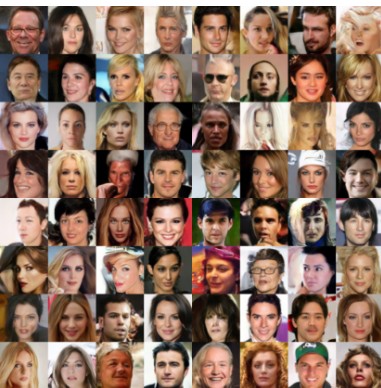

Figure 1: (Left) Schematic representation of CoVAE. The VAE-style model is trained to learn a time-dependent latent distribution, which transitions to Gaussian as time increases. With a loss similar to Consistency Training, the reconstruction at a smaller time steps is used as a target (therefore does not receive gradients, represented with the dashed line) for the prediction at the bigger time steps. (Right) 2-step uncurated samples from CoVAE trained on CelebA 64.

## 1 Introduction

Deep Generative Models (DGMs) are deep neural networks trained to generate samples from an unknown data distribution, generally by learning a mapping between samples from such a distribution and random noise. Variational Autoencoders (VAEs) (Kingma, 2013; Rezende et al., 2014) were among the first DGMs to scale to high-dimensional data by learning a mapping with a lower

dimensional latent space. However, despite the several improvements such as the integration with Normalizing Flows (Rezende & Mohamed, 2015; Kingma et al., 2016; Kobyzev et al., 2020; Papamakarios et al., 2021) and the use of hierarchical latents (Sønderby et al., 2016; Maaløe et al., 2019; Child, 2021; Vahdat & Kautz, 2020), VAEs remained inferior in terms of generative performances when compared to other DGMs such as Generative Adversarial Networks (Goodfellow et al., 2014; Karras et al., 2020b). More recently, Diffusion Models (DMs) (Sohl-Dickstein et al., 2015; Ho et al., 2020; Song et al., 2021b) have achieved impressive state-of-the-art generative results in domains such as images (Karras et al., 2022; 2024) and video (Ho et al., 2022; Ruhe et al., 2024), but are limited in efficiency as they require several function evaluations for generation. Many alternatives have been proposed to achieve comparable performances while reducing the computational requirements, such as distillation methods (Salimans & Ho, 2022) or few-step models like Consistency Models (CMs) (Song et al., 2023; Lu & Song, 2025), Shortcut Models (Frans et al., 2025), Inductive Model Matching (Zhou et al., 2025) and MeanFlow (Geng et al., 2025a). However, all these methods are constrained to work in the data space, making it hard to scale to high-dimensional data. Moreover, there are several works showing that high-dimensional data generally live in a lower-dimensional manifold (Pope et al., 2021; Brown et al., 2023; Stanczuk et al., 2024; Loaiza-Ganem et al., 2024; Ventura et al., 2025) verifying the so-called manifold hypothesis (Bengio et al., 2013) and suggesting the benefit of working with lower-dimensional representations. Therefore, it became common practice to employ a two-stage training procedure, where a VAE is pretrained to perform dimensionality reduction, and then a powerful prior model such as the ones mentioned above is trained on the learned latent space, allowing for efficient and scalable generation (Rombach et al., 2022; Podell et al., 2024; Esser et al., 2024).

In this work, we introduce a single-stage training procedure for generative autoencoders that combines $\beta$-VAEs and consistency model into a unified framework. To this end, we propose a model that integrates training techniques from discrete Consistency Models with a time-dependent VAE, which we name Consistency Training of Variational AutoEncoders (CoVAE) (see figure 1). Through a dedicated regularization scheme, the CoVAE encoder learns a sequence of progressively noised latent representation, which transitions from point masses to a standard Gaussian distribution. Each level of encoding correspond to a given value of $\beta$, making the approach similar to an amortized ensemble of $\beta$-VAEs. This resembles the forward processes commonly used in diffusion models but it is fully leaned through the encoder architecture and it performs meaningful feature disentanglement and compression similarly to a traditional VAE encoding. Trained by replacing the standard VAE reconstruction loss with a consistency reconstruction loss defined over the series of latents, CoVAE achieves high sample quality and diversity, dramatically improving over equivalent VAEs and approaching the performance of modern generative models.

## 2 BACKGROUND

### 2.1 VARIATIONAL AUTOENCODERS

In Variational Autoencoders (Kingma, 2013; Rezende et al., 2014), an encoder network $\mathcal{E}_{\phi}(.) : \mathbb{R}^D \to \mathbb{R}^{2d}$ parametrized by $\phi$ is used to learn a mapping for data points $x \sim p_{\text{data}}$ to a probability distribtion over a latent space $p(z \mid x)$. In the simplest case, this probability is assumed to be a diagonal Gaussian $q_{\phi}(z \mid x) = \mathcal{N}(\mathcal{E}_{\phi}^{\mu}(x), \mathcal{E}_{\phi}^{\sigma}(x)^2 I)$, with $\mathcal{E}_{\phi}^{\mu}$ and $\mathcal{E}_{\phi}^{\sigma}$ being partitions of the encoder output representing mean and standard deviation of the distribution. The encoder is paired with a prior distribution $p(z)$ over the latents, commonly spherical Gussian, and a decoder network $\mathcal{D}_{\theta}(.) : \mathbb{R}^d \to \mathbb{R}^D$ parametrized by $\theta$ is trained to map latents back to the data space. The architecture is trained with a variational loss, which can be expressed as:

$$\mathcal{L}_{\text{VAE}}(\theta, \phi, \beta) = \mathbb{E}_{x,z}\left[\|\mathcal{D}_{\theta}(z) - x\|^2 + \beta KL(\mathcal{N}(\mathcal{E}_{\phi}^{\mu}(x), \mathcal{E}_{\phi}^{\sigma}(x)^2 I)\|\mathcal{N}(0, I))\right], \quad (1)$$

where $\beta$ is a scalar hyperparameter that regulates the trade-off between deterministic decoding and posterior coverage.

The latent variable $z \sim q_{\phi}(z \mid x)$ is obtained with the so-called *reparametrization trick*, necessary for backpropagating the gradients to the encoder through the sampling operation:

$$z = \mathcal{E}_{\phi}^{\mu}(x) + \mathcal{E}_{\phi}^{\sigma}(x)\epsilon, \quad \epsilon \sim \mathcal{N}(0, I). \quad (2)$$

In this expression, the mean encoder $\mathcal{E}_\phi^\mu(x)$ defines the deterministic latent embedding of the data point $x$ while $\mathcal{E}_\phi^\sigma(x)$ regulates the level of additive white noise. The effective signal-to-noise ratio $\left\|\mathcal{E}_\phi^\mu(x)\right\|^2 / \left\|\mathcal{E}_\phi^\sigma(x)\right\|^2$ implicitly depends on $\beta$, with low values corresponding to near-deterministic latent encoding. In general, increasing $\beta$ promotes disentanglement in the latent space, while smaller values of $\beta$ favor reconstruction. However, while generally improving performance, it is hard to find a value of $\beta$ which perfectly recovers the prior while generating high-quality samples (Higgins et al., 2017; Burgess et al., 2018). While theoretically sound, VAEs are known to generate relatively poor-quality samples, mostly due to the *prior hole* problem (Hoffman & Johnson, 2016; Rosca et al., 2018), which happens when the aggregate posterior fails to match the prior, resulting in regions of the prior which are not decoded to in-distribution data. A widespread solution is to train with small $\beta$ for good reconstruction, and then train post-hoc a powerful generative model as prior distribution on the learned latent. While this effectively solves the prior hole problem, it results in additional training compute and model parameters, as well as increased sampling time.

## 2.2 DIFFUSION MODELS

Here, we will offer a minimalist introduction to diffusion models specifically designed to connect with related concepts and formulas in VAEs. For a complete SDE based formulation see Song et al. (2021b). A diffusion model is defined by its time-dependent noise-injection model, which usually has a linear Gaussian form:

$$\boldsymbol{x}_t = \mathcal{F}(\boldsymbol{x}, t) = a_t \boldsymbol{x} + b_t \boldsymbol{\epsilon}, \quad \boldsymbol{\epsilon} \sim \mathcal{N}(\mathbf{0}, \boldsymbol{I}), \tag{3}$$

where $a_t$ and $b_t$ are time dependent scalar functions. As $t$ increases, $\boldsymbol{x}_t$ becomes more heavily noised, until $\boldsymbol{x}_T \approx \mathcal{N}(\mathbf{0}, \boldsymbol{I})$ at the maximum time step $T$. Then, a time dependent neural network $\hat{\boldsymbol{x}}_\theta(.,.): \mathbb{R}^D \to \mathbb{R}^D$ is trained to predict the original clean sample $\boldsymbol{x}$ from its corrupted version $\boldsymbol{x}_t$:

$$\mathcal{L}_{\text{DSM}}(\boldsymbol{\theta}) = \mathbb{E}_t\left[\lambda(t)\mathbb{E}_{\boldsymbol{x}}\left[\mathbb{E}_{\boldsymbol{x}_t|\boldsymbol{x}}\left[\|\hat{\boldsymbol{x}}_{\boldsymbol{\theta}}(\boldsymbol{x}_t, t) - \boldsymbol{x}\|^2\right]\right]\right] \tag{4}$$

where $\lambda(t)$ is a time dependent weighting function. We refer to this objective as the Denoising Score Matching (DSM) loss. Once the model is trained, one may sample from $p_{\text{data}}$ by starting at pure noise $\boldsymbol{x}_T = \boldsymbol{\epsilon} \sim \mathcal{N}(\mathbf{0}, \boldsymbol{I})$ and integrating a deterministic dynamical system (Song et al., 2021a):

$$\boldsymbol{x}_{t-\Delta t} = \boldsymbol{x}_t - \Delta t \left(\dot{a}_t \hat{\boldsymbol{x}}_{\boldsymbol{\theta}}(\boldsymbol{x}_t, t) + \dot{b}_t \hat{\boldsymbol{\epsilon}}(\boldsymbol{x}_t, t)\right) \tag{5}$$

where $\hat{\boldsymbol{\epsilon}}$ is an estimate of the noise obtained from the model prediction:

$$\hat{\boldsymbol{\epsilon}}(\boldsymbol{x}_t, t) = \frac{\boldsymbol{x}_t - a_t \hat{\boldsymbol{x}}_{\boldsymbol{\theta}}(\boldsymbol{x}_t, t)}{b_t}, \tag{6}$$

In practice one can use a discretization schedule $t_0, t_1, \ldots, t_{N-1}$, set $\Delta t = t_i - t_{i-1}$, and iterate equation 5 until $t_0$, yielding a sample $\boldsymbol{x} \approx p_{\text{data}}$.

## 2.3 CONSISTENCY MODELS

Consistency Models (Song et al., 2023) are a recent alternative to DMs designed for one or few-step generation. In CMs, a time dependent neural network $\boldsymbol{f}_{\boldsymbol{\theta}}(.,.): \mathbb{R}^D \to \mathbb{R}^D$ is trained to learn the solution to the deterministic denoising process in Eq. 5 without the need for an explicit numerical integration. More specifically, CMs directly learn the mapping from $\boldsymbol{x}_t$ to $\boldsymbol{x}$ instead of learning the vector field that determines the dynamics like a diffusion model. CMs must satisfy two conditions, namely the boundary condition $\boldsymbol{f}_{\boldsymbol{\theta}}(\boldsymbol{x}, 0) = \boldsymbol{x}$ and the self-consistency condition $\boldsymbol{f}_{\boldsymbol{\theta}}(\boldsymbol{x}_t, t) = \boldsymbol{f}_{\boldsymbol{\theta}}(\boldsymbol{x}_{t'}, t')$, which states that points $\boldsymbol{x}_t$ and $\boldsymbol{x}_{t'}$ on the same deterministic denoising path at different time steps $t$ and $t'$ should map to the same solution. In practice, these conditions are commonly enforced with the preconditioning from Karras et al. (2022):

$$\boldsymbol{f}_{\boldsymbol{\theta}}(\boldsymbol{x}_t, t) = c_{\text{skip}}(t)\boldsymbol{x}_t + c_{\text{out}}(t)\boldsymbol{F}_{\boldsymbol{\theta}}(\boldsymbol{x}_t, t), \tag{7}$$

where $c_{\text{skip}}(.)$ and $c_{\text{out}}(.)$ are time dependent scalar functions such that $c_{\text{skip}}(0) = 1$ and $c_{\text{out}}(0) = 0$. While CMs can be trained with a continuous formulation, the continuous objective can be subject

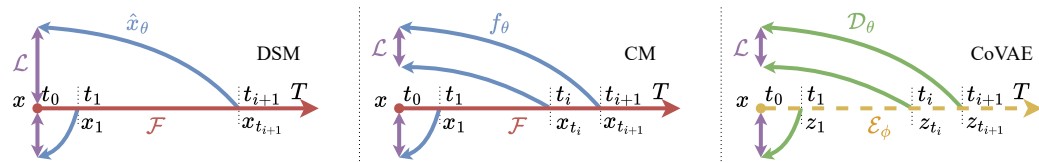

Figure 2: In this figure, we show a diagram of how CoVAE works compared to Diffusion and Consistency Models. In Diffusion and Consistency, a forward process $\mathcal{F}$ is used to add noise to data depending on the time step $t$. Then a network $\hat{\boldsymbol{x}}_{\boldsymbol{\theta}}$ is trained to be the average denoiser for DSM, or $\boldsymbol{f}_{\boldsymbol{\theta}}$ to match the prediction at the previous time step for discrete CMs with the reconstruction loss $\mathcal{L}$. In CoVAE, the encoder $\mathcal{E}_{\boldsymbol{\phi}}$ is used to obtain a noisy latent state $\boldsymbol{z}_t$ with the reparametrization trick, similarly to $\mathcal{F}$. However, in this case we use a dashed line, as the noising process is enforced by the regularization term $\beta(t)$, and there is not a direct relationship with time. The decoder $\mathcal{D}_{\boldsymbol{\theta}}$ then maps the latents back to the data space, and is trained with a loss similar to CMs.

to instabilities and requires several technicalities to work properly (Lu & Song, 2025). A common alternative is the discrete time objective:

$$\mathcal{L}_{\text{CM}}^{\text{disc}}(\boldsymbol{\theta}) = \mathbb{E}_{\boldsymbol{x}_t, t}\left[\lambda(t)\,\|\boldsymbol{f}_{\boldsymbol{\theta}}(\boldsymbol{x}_t, t) - \boldsymbol{f}_{\boldsymbol{\theta}^-}(\boldsymbol{x}_{t'}, t')\|^2\right] \tag{8}$$

where $t$ and $t'$ are neighboring time steps chosen according to the defined discretization strategy (Song & Dhariwal, 2024; Geng et al., 2025b), and $\boldsymbol{\theta}^-$ is a frozen copy of the network parameters which does not require gradients. Intuitively, this loss function gradually "bootstraps" the initial boundary condition at $t = 0$ to the final pure noise state by minimizing differences along the path. During training, the discretization steps transition from coarse to fine grained, bootstrapping the signal from early time steps to the later ones. Sampling from CMs can be then done in a single or few steps by predicting the initial conditions of samples from the final noise distribution. For our current purposes, it is important to note that, due to the boundary conditions, the consistency loss reduces to a conventional autoencoder reconstruction loss when $t' = 0$:

$$\|\boldsymbol{f}_{\boldsymbol{\theta}}(\boldsymbol{x}_t, t) - \boldsymbol{x}\|^2 \tag{9}$$

where $\boldsymbol{x}_t = a_t\boldsymbol{x} + b_t\boldsymbol{\epsilon}$. This suggests that the network $\boldsymbol{f}_{\boldsymbol{\theta}}(\boldsymbol{x}_t, t)$ can be interpreted as a decoder architecture and that the forward noise-injection model $\mathcal{F}(\boldsymbol{x}, t)$ can be interpreted as a non-learned (and rather trivial) encoder.

## 3 METHOD

Before introducing our contribution, we highlight some similarities between the models discussed in section 2, also shown in figure 2.

We first direct the attention to equations 2 and 3, noting that the forward kernel commonly used in diffusion can be seen as a time-dependent version of the reparametrization trick, where drift and diffusion terms are simple, predefined, dimensionality-preserving transformations. On the contrary, VAEs use learned nonlinear dimensionality-reducing mapping as drift and diffusion terms. Note that in DMs the diffusion term does not need to be scalar and can be data dependent (see appendix A from Song et al. (2021b)).

To bridge the gap between VAEs and DMs, one can extend $\beta$-VAEs to learn a time dependent encoding and decoding:

$$\mathcal{L}_{t-\text{VAE}}(\boldsymbol{\theta}, \boldsymbol{\phi}) = \mathbb{E}_t[\mathcal{L}_{\text{VAE}}(\theta, \phi, \beta(t))], \tag{10}$$

where $\beta(t)$ is a monotonically increasing weighting function. By appropriately defining $\beta(t)$, the learned latents transition between delta distributions and Gaussians, and the reparametrization trick can be expressed as:

$$\boldsymbol{z}_t = \mathcal{E}_{\boldsymbol{\phi}}^{\mu}(\boldsymbol{x}, t) + \mathcal{E}_{\boldsymbol{\phi}}^{\sigma}(\boldsymbol{x}, t)\boldsymbol{\epsilon}, \quad \boldsymbol{\epsilon} \sim \mathcal{N}(\mathbf{0}, \boldsymbol{I}). \tag{11}$$

Equation 11 now resembles closely a forward noise-injection model, effectively defining an underlying latent noising process. The reconstruction part of equation 10 is now the latent-to-data equivalent

to the DSM loss from equation 4. In this form, the resulting model would not bring any tangible benefit over standard $\beta$-VAEs, as it would simply result in a amortization of several $\beta$-VAEs, sharing the same limitations of each single model, i.e. the limiting tradeoff between reconstruction and generation quality.

## 3.1 TRAINING VAES AS CONSISTENCY MODELS

Our main proposal is to replace the reconstruction loss of the time-dependent $\beta$-VAE from Eq. 10 with a latent consistency loss inspired by Eq. 8. Given a time discretization $t_0, t_1, \ldots, t_{N-1} = T$ with $N$ time steps, and imposing the identity function at $t_0$ as boundary condition, we can now define a loss for Consistency Training of Variational AutoEncoders (CoVAEs):

$$\mathcal{L}_{\text{CoVAE}}(\boldsymbol{\theta}, \boldsymbol{\phi}) = \mathbb{E}_{\boldsymbol{x}, \boldsymbol{z}, t_i} \left[ \lambda(t_i) \left\| \boldsymbol{\mathcal{D}}_{\boldsymbol{\theta}}(\boldsymbol{z}_{t_i}, t_i) - \boldsymbol{\mathcal{D}}_{\boldsymbol{\theta}^-}(\boldsymbol{z}_{t_{i-1}}, t_{i-1}) \right\|^2 \right. \tag{12}$$
$$\left. + \beta(t_i) KL(\mathcal{N}(\boldsymbol{\mathcal{E}}_{\boldsymbol{\phi}}^\mu(\boldsymbol{x}, t_i), \boldsymbol{\mathcal{E}}_{\boldsymbol{\phi}}^\sigma(\boldsymbol{x}, t_i)^2 \boldsymbol{I}) \| \mathcal{N}(\boldsymbol{0}, \boldsymbol{I})) \right],$$

where $\lambda(t)$ is a monotonically decreasing weighting function generally used in CMs, and $\boldsymbol{\theta}^-$ are copies of the model parameters that do not receive gradients. The latent variables $\boldsymbol{z}_{t_i}$ and $\boldsymbol{z}_{t_{i-1}}$ are obtained with the time-dependent reparametrization trick from equation 11 using the same random direction $\boldsymbol{\epsilon}$ at both times $t_i$ and $t_{i-1}$. With the CoVAE objective, the decoder learns the solution of the latent dynamics and the latent-to-data mapping jointly. Like in discrete CMs, such a solution is bootstrapped from the earlier time steps to the later ones. The resulting model is a autoencoder that can be trained end-to-end with the Consistency Training objective from equation 12. Note that, differently from DMs and CMs, where the time has a direct effect on the forward process, for CoVAE the effect of time is implicitly defined by the weighting functions $\beta(t)$ and $\lambda(t)$, which regulate the level of feature disentanglement.

## 3.2 BOUNDARY CONDITIONS IN LATENT SPACE

While imposing the identity function at $t_0$ is enough to respect the initial condition required by CMs, we found in practice that using such a simple parametrization can lead to instabilities during training (see Appendix C.1). Similarly to CMs, we aim to incorporate an EDM-style parametrization from equation 7, which cannot be directly used in our settings, as the latent variable $\boldsymbol{z}_t$ and the output of the decoder generally live in spaces with different dimensionality. We propose a different parametrization where instead of using the noisy state $\boldsymbol{x}_t$ (or latent state $\boldsymbol{z}_t$ in CoVAE), we use a learned approximation of the average decoder $\mathbb{E}[\boldsymbol{x} \mid \boldsymbol{z}_t]$. The average decoder will be a faithful reconstruction for small $t$, while will be a blurry reconstruction as $t$ increases. This quantity is unknown but can be obtained by training a neural network $\hat{\boldsymbol{x}}_{\boldsymbol{\theta}}$ with the time-dependent VAE loss from equation 10. Note that we use on purpose the same notation $\hat{\boldsymbol{x}}_{\boldsymbol{\theta}}$ as for DSM, to stress that the role of the network is the same, i.e. to recover the clean data from the latent $\boldsymbol{z}_t$ or noisy state $\boldsymbol{x}_t$. The decoder parametrization becomes as follows:

$$\boldsymbol{\mathcal{D}}_{\boldsymbol{\theta}}(\boldsymbol{z}_t, t) = c_{\text{skip}}(t)\hat{\boldsymbol{x}}_{\boldsymbol{\theta}^-}(\boldsymbol{z}_t, t) + c_{\text{out}}(t)\boldsymbol{r}_{\boldsymbol{\theta}}(\boldsymbol{z}_t, t), \tag{13}$$

where $\boldsymbol{r}_{\boldsymbol{\theta}}$ models the residual of the average decoder network. Note how the parameters of the average decoder network are frozen and do not receive gradients. Instead, the reconstruction part of the CoVAE loss is modified with an additional denoiser-style loss:

$$\mathcal{L}_{\text{CoVAE}}^{\text{rec}} = \mathbb{E}_{\boldsymbol{x}, \boldsymbol{z}, t_i} \left[ \lambda(t_i) \left( \left\| \boldsymbol{\mathcal{D}}_{\boldsymbol{\theta}}(\boldsymbol{z}_{t_i}, t_i) - \boldsymbol{\mathcal{D}}_{\boldsymbol{\theta}^-}(\boldsymbol{z}_{t_{i-1}}, t_{i-1}) \right\|^2 + \lambda_{\text{d}}(t) \left\| \hat{\boldsymbol{x}}_{\boldsymbol{\theta}}(\boldsymbol{z}_{t_i}, t_i) - \boldsymbol{x} \right\|^2 \right) \right], \tag{14}$$

where $\lambda_{\text{d}}(.)$ is another time dependent weighting function used to regulate the interplay between consistency and denoising loss. In practice, we double the output channels of the decoder, and use half for $\hat{\boldsymbol{x}}_{\boldsymbol{\theta}}$ and the other half for $\boldsymbol{r}_{\boldsymbol{\theta}}$, resulting in a negligible increase in model parameters and compute. This is motivated by the fact that the weights of the denoiser network are generally used as initialization for training CMs (Geng et al., 2025b; Lu & Song, 2025), which suggests a certain degree of compatibility between the features needed for the two losses. The identity function at $t_0$ is still applied.

---

**Algorithm 1** CoVAE Loss

---

**Input:** data distribution $p_{\text{data}}$, decoder parameters $\boldsymbol{\theta}$, encoder parameters $\boldsymbol{\phi}$, weighting functions $\lambda(\cdot)$, $\beta(\cdot)$ and $\lambda_{\text{d}}(.)$, discrete time step distribution $p(t)$

Sample $\boldsymbol{x} \sim p_{\text{data}}$, $t_i \sim p(t)$, $\boldsymbol{\epsilon} \sim N(\mathbf{0}, \boldsymbol{I})$

$\boldsymbol{z}_{t_i} \leftarrow \mathcal{E}_{\boldsymbol{\phi}}^{\boldsymbol{\mu}}(\boldsymbol{x}, t_i) + \mathcal{E}_{\boldsymbol{\phi}}^{\boldsymbol{\sigma}}(\boldsymbol{x}, t_i)\boldsymbol{\epsilon}$

$\boldsymbol{z}_{t_{i-1}} \leftarrow \mathcal{E}_{\boldsymbol{\phi}^-}^{\boldsymbol{\mu}}(\boldsymbol{x}, t_{i-1}) + \mathcal{E}_{\boldsymbol{\phi}^-}^{\boldsymbol{\sigma}}(\boldsymbol{x}, t_{i-1})\boldsymbol{\epsilon}$

$\mathcal{L}_{\text{CoVAE}}^{\text{d}}(\boldsymbol{\theta}, \boldsymbol{\phi}) \leftarrow \|\hat{\boldsymbol{x}}_{\boldsymbol{\theta}}(\boldsymbol{z}_t, t) - \boldsymbol{x}\|^2$

$\mathcal{L}_{\text{CoVAE}}^{\text{cm}}(\boldsymbol{\theta}, \boldsymbol{\phi}) \leftarrow \left\|\mathcal{D}_{\boldsymbol{\theta}}(\boldsymbol{z}_t, t), \mathcal{D}_{\boldsymbol{\theta}^-}(\boldsymbol{z}_{t_{i-1}}, t_{i-1})\right\|^2$

$\mathcal{L}_{\text{CoVAE}}^{\text{kl}}(\boldsymbol{\phi}) \leftarrow KL(\mathcal{N}(\mathcal{E}_{\boldsymbol{\phi}}^{\boldsymbol{\mu}}(\boldsymbol{x}, t_i), \mathcal{E}_{\boldsymbol{\phi}}^{\boldsymbol{\sigma}}(\boldsymbol{x}, t_i)^2\boldsymbol{I})\|\mathcal{N}(\mathbf{0}, \boldsymbol{I}))$

$\mathcal{L}_{\text{CoVAE}}(\boldsymbol{\theta}, \boldsymbol{\phi}) \leftarrow \lambda(t_i)[\mathcal{L}_{\text{CoVAE}}^{\text{cm}}(\boldsymbol{\theta}, \boldsymbol{\phi}) + \lambda_{\text{d}}(t_i)\mathcal{L}_{\text{CoVAE}}^{\text{d}}(\boldsymbol{\theta}, \boldsymbol{\phi})] + \beta(t_i)\mathcal{L}_{\text{CoVAE}}^{\text{kl}}(\boldsymbol{\phi})$

**Output:** $\mathcal{L}_{\text{CoVAE}}(\boldsymbol{\theta}, \boldsymbol{\phi})$

---

### 3.3 Training and sampling with CoVAE

We report the pseudocode for the CoVAE loss in algorithm 1. After training, the model can be used to generate data by decoding samples from the prior like in stadard VAEs. In addition, CoVAEs can leverage the learned time dependent latent mapping to perform multi-step sampling, similarly to CMs, by re-encoding the generated data at intermediate time steps, adding new noise with the reparametrization trick, and re-denoising (see algorithm 2). Note that, for a two-step sampling procedure, three function evaluations are required (twice the decoder and once the encoder). We report the several design choices used for training CoVAE in Appendix B, while we describe alternative formulations for CoVAE in Appendix E.

---

**Algorithm 2** Multistep CoVAE Sampling

---

**Input:** Decoder $\mathcal{D}_{\boldsymbol{\theta}}$, encoder $\mathcal{E}_{\boldsymbol{\phi}}$, sequence of time points $\tau_1 > \tau_2 > \cdots > \tau_{N-1}$

Sample $\boldsymbol{\epsilon} \sim \mathcal{N}(\mathbf{0}, \boldsymbol{I})$

$\boldsymbol{x} \leftarrow \mathcal{D}_{\boldsymbol{\theta}}(\boldsymbol{\epsilon}, \tau_1)$ $\qquad\qquad\qquad\qquad\qquad\qquad\qquad$ ▷ Choose $\tau_1 = \sigma_{\max}$

**for** $n = 2$ **to** $N - 1$ **do**

$\qquad$ Sample $\boldsymbol{\epsilon} \sim \mathcal{N}(\mathbf{0}, \boldsymbol{I})$

$\qquad \boldsymbol{z}_{\tau_n} \leftarrow \mathcal{E}_{\boldsymbol{\phi}}^{\boldsymbol{\mu}}(\boldsymbol{x}, \tau_n) + \mathcal{E}_{\boldsymbol{\phi}}^{\boldsymbol{\sigma}}(\boldsymbol{x}, \tau_n)\boldsymbol{\epsilon}$

$\qquad \boldsymbol{x} \leftarrow \mathcal{D}_{\boldsymbol{\theta}}(\boldsymbol{z}_{\tau_n}, \tau_n)$

**end for**

**Output:** $\boldsymbol{x}$

---

## 4 Experiments

In this section, we report experimental results on common image benchmarks. We use Frechet Inception Distance (FID) (Heusel et al., 2017) as an evaluation metric, both on $50k$ samples from the models and on encoded-decoded training images using the whole dataset, to evaluate generative and reconstruction performance. For CoVAE, we always use $t = \sigma_{\min}$ to compute the reconstruction FID. We provide additional visualizations and samples from our models in Appendix D.

### 4.1 MNIST

As a simple benchmark, we compare the results from an equivalent VAE, $\beta$-VAE and CoVAE on MNIST (Deng, 2012), where we train a model with a $7 \times 7$ latent space and one channel ($16\times$ compression rate). The models were trained for $400k$ iterations with batch size 128 and EMA rate 0.9999. CoVAE was trained with the hyperparameters described in Appendix B. From the results in table 1, we can see how CoVAE shows significantly improved results without having to trade-off generation and reconstruction like in $\beta$-VAEs, confirming the benefits of the bootsrtapped

---

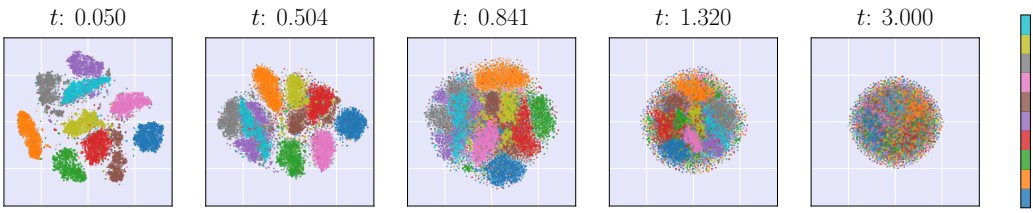

Figure 3: t-SNE embedding of samples from the encoded latents for $10k$ MNIST images at different time steps, with consistent per-sample noise mask across the time steps $t$.

time-dependent objective. To further analyze the behavior of the learned time-dependent latent representation, we show in figure 3 the result of 2D t-SNE (Van der Maaten & Hinton, 2008) on samples from the embedding of $10k$ images from the trainig set for different time steps, with the same noise mask used for sampling across different time steps. From small time steps, the samples from each class are embedded in well separated areas, while they gradually become more random as time increases. Additional latent space visualizations are reported in Appendix D.1, figure 7. Similarly, in figure 4 we show the Signal-to-Noise Ratio (SRN) of the learned latent space averaged over the same $10k$ image embeddings. From the plot, we can see how the average SNR transitions from a very large value corresponding to almost no noise, to a value close to zero approaching pure noise in latent space.

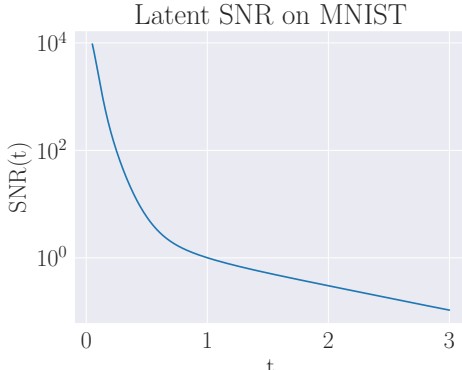

Figure 4: SNR of the latent space for the trained CoVAE model over the different time steps.

| FID ($\downarrow$) on MNIST | | | |
|---|---|---|---|
| | 1 step | 2 steps | Rec. |
| VAE | 17.2 | - | 21.17 |
| $\beta$-VAE ($\beta = 0.5$) | 13.24 | - | 16.56 |
| CoVAE (ours) | **5.62** | **3.83** | **2.19** |

Table 1: FID results (lower is better) for Co-VAE and VAE baselines on MNIST.

## 4.2 CIFAR-10

We use CIFAR-10 (Krizhevsky et al., 2009) to assess the generative peroformance of CoVAE. as it is a common image benchmark for DGMs. We refer the reader to Appendix B for a description of all the hyperparameters used, and to Appendix C.1 for an ablation over different configurations used for CoVAE, including a comparison with and without the boundary conditions from equation 13. In the following, we train our models using the 112M parameters configuration (meaning that the decoder has roughly the same number of parameters as the architectures commonly used in CMs for CIFAR-10) and batch size 1024. For the baselines and CoVAE, we use a 1024-dimensioanl latent space. To further improve the generative performance, we train CoVAE with a patch-based adversarial loss $\mathcal{L}_{\text{adv}}$ like in Esser et al. (2021); Rombach et al. (2022). In table 2 we report the results obtained with our model, compared to the VAE and $\beta$-VAE baselines, as well as NVAE (Vahdat & Kautz, 2020) and DC-VAE (Parmar et al., 2021). These baselines were chosen as they are the best performing VAE-based methods using a single stage training procedures and without training a separate prior to sample from. See Appendix A for a more detailed discussion about the baselines, and Appendix C.2 for a comparison with a broader selection of models. CoVAE

significantly outperforms the equivalent VAE and $\beta$-VAE baselines, and outperforms both NVAE and DC-VAE, and the additional adversarial loss further improves generative and reconstruction performance. Two-steps samples from CoVAE w/ $\mathcal{L}_{\text{adv}}$ are shown in figure 5.

| **FID ($\downarrow$) on CIFAR-10** | | | |
|---|---|---|---|
| Model | 1 step | 2 steps | Rec. |
| VAE | 96.09 | - | 60.76 |
| $\beta$-VAE ($\beta = 0.1$) | 66.79 | - | 30.23 |
| NVAE | 23.49 | - | 2.67 |
| DC-VAE | 17.9 | - | 21.4 |
| CoVAE (ours) | 17.21 | 14.06 | 2.36 |
| CoVAE w/ $\mathcal{L}_{\text{adv}}$ (ours) | **11.69** | **9.82** | **2.15** |

| **FID ($\downarrow$) on CelebA 64** | | | |
|---|---|---|---|
| Model | 1 step | 2 steps | Rec. |
| NVAE | 14.74 | - | - |
| DC-VAE | 19.9* | - | 14.3* |
| CoVAE (ours) | 10.4 | 9.4 | 5.67 |
| CoVAE w/ $\mathcal{L}_{\text{adv}}$ (ours) | **8.27** | **7.15** | **4.90** |

*DC-VAE reports results on $128 \times 128$ resolution.

Table 2: FID ($\downarrow$) on CIFAR-10 (left) and CelebA 64 (right). Lower is better. "Rec." is Reconstruction FID.

### 4.3 CelebA 64 and image manipulation

We further test CoVAE on CelebA Liu et al. (2015) resized to $64 \times 64$, as it is another common baseline for VAE-based methods. Also here we use CoVAE with $\times 3$ compression rate, corresponding of latent space of 4096 dimensions. We report the FID results in table 2. CoVAE achieves high sample quality and reconstruction compared to the baselines, with samples shown in figure 1. Similarly to other VAE-based models, the latent space learned by CoVAE can be used for image manipulation. However, in our case we have access to a time-dependent latent space, which allows to trade-off between faithful reconstruction and disentanglement in latent space. We show the effect of latent space interpolation at different time steps in figure 5, and provide a comprehensive analysis in Appendix D.

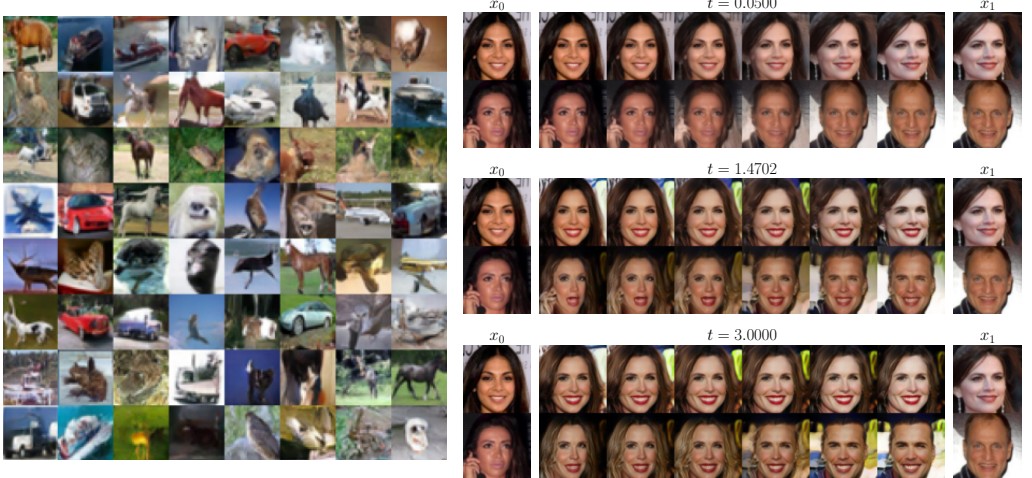

Figure 5: (Left) 2-step samples from CIFAR-10 with CoVAE w/ $\mathcal{L}_{\text{adv}}$. (Right) Latent interpolation on CelebA with different interpolation strengths.

## 5 Related Work

**Single and few-steps generative models**: Recently, many single or few-step generative models that can retain the generative performance of diffusion models have emerged, such as Consistency Models (Song et al., 2023; Song & Dhariwal, 2024; Geng et al., 2025b; Lu & Song, 2025; Dao et al., 2025), Shortcut Models (Frans et al., 2025), Inductive Model Matching (Zhou et al., 2025) and MeanFlow (Geng et al., 2025a), and have also been used as priors for pretrained VAEs, further

reducing sampling time while retaining comparable generative performance. With our method, we aim to show that despite the success obtained by two-stage training procedures, it is possible to obtain a competitive VAE with a single training stage and simple prior and posterior distributions. To an extent, CoVAE can be seen as a way to unify latent CMs and pretrained VAE into a single model.

**Learning the forward process**: Several works have explored learning data-dependent forward processes in ambient space: Nielsen et al. (2024) use an encoder to parameterize the noise injection, Bartosh et al. (2024a) learn the drift and diffusion terms of the SDE directly, while Bartosh et al. (2024b) further extend this by combining invertible flows with diffusion for exact likelihoods. Unlike these methods, which still rely on separate score models or iterative sampling in data space, CoVAE learns a progressive noising process directly in latent space and unifies encoding, noising, and decoding within a single time-dependent VAE trained with a consistency objective. Other methods (Pooladian et al., 2023; Liu et al., 2023; Lee et al., 2023; Albergo et al., 2024; Li et al., 2024; Silvestri et al., 2025) implicitly alter the forward process by introducing a coupling between data and noise. Among these, Lee et al. (2023); Silvestri et al. (2025) use a neural network to learn the data-noise coupling in a VAE-style formulation, showing similarities to CoVAE, especially Silvestri et al. (2025) which applies this idea in the context of CMs. However, all these methods remain restricted to the ambient space, while CoVAE jointly learns the latent mapping and the forward process directly in latent space, including the latent-to-noise coupling.

**Using a time-depedent VAE**: Some recent works employ a time-dependent VAE architecture similar to ours. Specifically, Batzolis et al. (2023) uses a time-dependent encoder in combination with a pretrained score model which can be directly used as a decoder, effectively obtaining an improved VAE method, but still requiring the iterative sampling procedure of DMs. The work from Uppal et al. (2025) also uses a time dependent $\beta$-VAE regularized to obtain a latent space that transitions to isotropic Gaussian as time increases. However, they then need to train a non-linear diffusion model in such a latent space, falling in the two-stage training procedures, and require several steps for sampling. Compared to these methods, we solely rely on the time-dependent VAE, and can perform generation in one or few steps.

## 6 CONCLUSIONS

In this work, we have introduced CoVAE, a unified, single-stage training framework that combines VAEs with a consistency-based decoder loss to enable high-quality one- or few-step sampling, without resourcing to complex priors or multi-stage training. We further provided a set of design choices for training, achieving high sample quality on image generation benchmarks.

**Limitations and future work**: While CoVAE shows promising results, we highlight here the main limitations and possible directions for further improvement. A limitation compared to VAEs is that with our formulation we cannot easily compute a tight evidence lower bound, making it non-trivial to evaluate the data log-likelihood.

The proposed training strategy was mainly driven by empirical evaluations, and relies on hyper-parameters such as weighting functions and discretization scheme. Deriving more principled design choices could lead to a simpler training procedure as well as improved performance. The chosen architecture is similar to the one used in latent diffusion (Rombach et al., 2022), but recent works (Skorokhodov et al., 2025; Chen et al., 2025) show how improving the architecture leads to substantial boost in generative performance, efficiency, and compressibility. We believe that designing an ad-hoc architecture for CoVAE could bring similar benefits. Finally, there have been many improvements in the training techniques of Consistency Models, and integrating some strategies with CoVAE such as initializing from a pretrained model (Geng et al., 2025b), focusing the training on later time steps (Lee et al., 2025), and using a continuous formulation (Lu & Song, 2025) could lead to superior results. The close structural similarities between CoVAEs and VAEs suggest that performance can also be improved using well known VAE approaches such as the use of structured prior and posterior models. In particular, hierarchical approaches have proven to be very effective in VAEs (Vahdat & Kautz, 2020) and could lead to substantial improvements in CoVAE generative performance. Note that any pre-existing VAE architecture can be extended for CoVAE training by simply adding time conditioning.

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

## A   A BROADER DISCUSSION ON COVAE AND RELATED METHODS

In this section, we provide a comprehensive comparison between CoVAE and other related methods in the literature. The main objective of this work was to develop a competitive generative model, that could generate samples with one or few steps while trained with an end-to-end procedure and with latent spaces of arbitrary dimensions. At its core, CoVAE is an extension of $\beta$-VAEs, by learning several latent representations with increasing disentanglement, and using the bootstrapped reconstruction loss from CMs. From our experiments, it is clear how our formulation is superior to VAEs and $\beta$-VAEs with equivalent architecture. As additional baselines, we have included NVAE (Vahdat

& Kautz, 2020), which is a powerful hierarchical VAE using a complex architecture including several normalizing flows. Due to the hierarchical nature, NVAE's latent dimensionality is generally much larger than the ambient space. CoVAE performs significantly better than NVAE, despite the simpler architecture and smaller latent dimensionality. We believe that the CM-like reconstruction loss could be beneficial to NVAE and similar hierarchical models, but we leave this exploration for future work. Another baseline included in our evaluation is DC-VAE (Parmar et al., 2021), as it uses a one-stage training procedure and a simple prior. Differently to our approach, DC-VAE can achieve great results with a much smaller latent representation (128 for CIFAR-10, 512 for CelebA) thanks to the combination of contrastive and adversarial losses. However, CoVAE can outperform DC-VAE even without adversarial loss, and the reconstruction quality obtained by DC-VAE is generally poor.

Regarding the performance comparison between two-stage latent models and CoVAE, we could not find many baselines using our same datasets. However, there are a few methods that report results for CIFAR-10 and CelebA 64, namely LSGM (Vahdat et al., 2021), VAEBM (Xiao et al., 2021), D2C (Sinha et al., 2021) and DiffuseVAE (Pandey et al., 2022). Notably, LSGM is also a single-stage training procedure, even though it still uses separate models for autoencoder and diffusion prior. They achieve an FID of 2.10 on CIFAR-10 with 138 function evaluations. VAEBM is a two stage training procedure combining NVAE with an energy based model prior. At inference, they to run an MCMC chain to sample from the prior. They achieve 12.19 FID on CIFAR-10 and 5.31 FID on CelebA 64. D2C combines a VAE with a diffusion prior and a contrastive loss. They achieve 10.11 FID on CIFAR-10 with 59 function evaluations, and 5.7 FID on CelebA 64 with 100 NFE. Finally, DiffuseVAE is also a latent diffusion model with a diffusion prior, and achieves 2.62 FID on CIFAR-10 and 3.97 on CelebA 64, with 1000 sampling steps from the prior. In comparison CoVAE still outperforms some of these models on CIFAR-10, and on CelebA 64 can perform comparably to VAEBM and D2C while requiring significantly less computation.

Compared to single stage models for few-steps sampling like Consistency Models, Shortcut Models, Inductive Model Matching, and MeanFlows, CoVAE achieves worse FID results. However, the aforementioned models are restricted to work in ambient space, making it hard scale to high-dimensional datasets unless used in combination with a pretrained autoencoder. The disadvantages of the two-stage models over CoVAE is the need for more training budget and memory (VAE model + prior model), and, perhaps less relevant, the need for more sampling steps. In fact, even for one step latent CMs, generating samples requires one forward from the prior and one from the decoder, while CoVAE requires only one decoder pass. However, for two-steps sampling, CoVAE requires three function evaluations, decoder, encoder, and decoder again, while latent CM requires two forward passes from the prior and one from the decoder. Whether or not it is an advantage to use CoVAE in this scenario depends on the specific architectural choices and the quality of the resulting samples. Similarly, training the VAE for latent models is generally faster than training CoVAE, so the one stage training procedure is advantageous only when training CoVAE is faster than training VAE + prior, which can become challenging to achieve for high-dimensional data.

## B    EXPERIMENTAL DETAILS

### B.1    DESIGN CHOICES FOR TRAINING COVAE

Training CoVAE requires a number of design choices which we discuss here. Similarly to discrete CMs, we have to choose discretization strategy, timestep distribution, weighting functions $\lambda(t)$, $\beta(t)$ and $\lambda_{\mathrm{d}}(t)$, the scalar functions $c_{\mathrm{in}}(t)$, $c_{\mathrm{skip}}(t)$ and $c_{\mathrm{out}}(t)$, and minimum and maximum time values $\sigma_{\min}$ and $\sigma_{\max}$. For the discretization strategy, we reuse the discretization introduced in Karras et al. (2022) and used in Song et al. (2023) for consistency training:

$$t_i = \left( \sigma_{\min}^{1/\rho} + \frac{i-1}{N(k)-1} \left( \sigma_{\max}^{1/\rho} - \sigma_{\min}^{1/\rho} \right) \right)^{\rho},  \tag{15}$$

where $\rho$ is a scalar hyperparameter controlling the "linearity" of the discretization ($\rho = 1$ results in a linear discretization, while increasing $\rho$ transitions towards logarithmic), $k \in [0, K]$ is the current training iteration, $K$ is the total training iterations, $i \in [1, N(k)]$ is the discretization step and $N(k)$ is a discretization curriculum returning the number of discretization steps at the current iteration. As

$N(k)$ we choose to use the exponential curriculum from Song & Dhariwal (2024):

$$N(k) = \min\left(s_0 2^{\lfloor \frac{k}{K'} \rfloor}, s_1\right) + 1, \quad K' = \left\lfloor \frac{K}{\log_2 \lfloor s_1/s_0 \rfloor + 1} \right\rfloor, \tag{16}$$

where $s_0 = 2$ and $s_1 = 256$ are initial and final number of steps respectively. During training, we sample time steps uniformly from the given discretization. In CMs, $t_1 = \sigma_{\min}$ is the minimum value that the time steps can assume, and the boundary conditions impose the identity at $t_1$. In CoVAE, we additionally add $t_0 = 0$ to the time steps and apply the boundary condition at $t_0$. This allows us to choose exactly $\sigma_{\min}$ as the first time step used by the encoder (while in CMs $\sigma_{\min}$ is never actually used as the boundary condition applies). After an initial tuning phase (details in B.4), we choose $\sigma_{\min} = 0.05$, $\sigma_{\max} = 3$, $\rho = 7$, $\beta(t) = t^2$ and $\lambda(t) = 1/t$. We always set $c_{\text{skip}}(t) = 1$ (using the derivation in section E.1.3 as a guideline), $c_{\text{in}}(t) = 1$, while for $c_{\text{out}}(t)$ and $\lambda_{\text{d}}(t)$, we use linear interpolations:

$$c_{\text{out}}(t) = \frac{t - \sigma_{\min}}{\sigma_{\max} - \sigma_{\min}}, \quad \lambda_{\text{d}}(t) = c_{\text{d}} + (1 - c_{\text{d}})\left(1 - \frac{t - \sigma_{\min}}{\sigma_{\max} - \sigma_{\min}}\right), \tag{17}$$

where $c_{\text{d}} = 0.1$ is used to reduce the effect of the average decoder loss as $t$ increases, to avoid conflict with the consistency loss. These linear weights are chosen empirically as we do not have an analytic expression for the SNR. The time $t$ is processed as $\log(t)/4$ like in Karras et al. (2022). For the decoder loss, we use the pseudo-huber loss defined like in Song & Dhariwal (2024), while for the average decoder we use the L2 loss. We train all the models for $400k$ iterations with Exponential Moving Average of the weights (EMA), with rate $\mu_{\text{EMA}} = 0.9999$. During training, we make sure that the network uses the same dropout mask for target and prediction computation, as commonly done in CMs. As architecture, we reuse DDPM++ (Ho et al., 2020; Karras et al., 2022) which is based on U-Net (Ronneberger et al., 2015), but without the skip connections between different latent resolutions, effectively obtaining a non-hierarchical time-dependent VAE architecture, where the latent dimensionality is defined by the spatial resolution and channels after encoding. We reuse also the middle block and $1 \times 1$ convolution from Rombach et al. (2022). Following Lu & Song (2025), we replace the Adaptive Group Normalization with Adaptive Double Normalization. The latent size is defined by an the number of channels $z_{\text{ch}}$ multiplied by the spatial resolution after the encoder. The encoder reduces spatial dimensionality by 8 for MNIST and CIFAR-10, and by 16 for CelebA 64.

## B.2 ADVERSARIAL LOSS

The adversarial loss is used only after $k_w$ warm-up steps, which we set to be half of the training iterations, and is multiplied by a scaling factor:

$$\lambda_{\text{adv}}(k, t) = \lambda(t)\frac{k - k_w}{K - k_w} * \lambda_{\text{adv}} * \mathbb{I}\left(t > \frac{\sigma_{\max}(k - k_w)}{K - k_w}\right), \tag{18}$$

where $k$ is the current training iteration, $\lambda(t)$ is the same time weighting function used for the CoVAE loss, $\lambda_{\text{adv}} = 0.05$ is a constant hyperparameter, and $\mathbb{I}(.)$ is a gate function which applies the adversarial loss to time steps progressively as the iterations increase. The rationale behind the gating is that smaller time steps are better approximations of the data earlier during training.

## B.3 NETWORK CONFIGURATIONS

In section C.1 we use different configurations for the neural network in the ablation for different network size. The network differ for the channel multipliers and number of residual blocks as follows:

- Model with 35.8M parameters: Channel multiplyers $= [2, 2, 2]$, resdual blocks $= 2$;
- Model with 54.2M parameters: Channel multiplyers $= [2, 2, 2]$, resdual blocks $= 4$;
- Model with 94M parameters: Channel multiplyers $= [2, 2, 4]$, resdual blocks $= 2$.

We report in table 3 the hyperparameters used for the models in section 4. All our models are trained with precision BFloat16. For training, we use the random seed 42, while for evaluation we set it to 32.

| Model Setups | MNIST | CIFAR-10 | CelebA 64 |
|---|---|---|---|
| Model Channels | 64 | 128 | 128 |
| N° of ResBlocks | 2 | 4 | 2 |
| Attention Resolution | [14] | [16,8] | [16,8] |
| Channel multiplyer | [2, 2, 2] | [2, 2, 4] | [1, 2, 2, 4] |
| Model capacity | 8M | 112M | 81.7M |
| Latent size | 49 | 1024 | 4096 |
| Discriminator Capacity | - | 3M | 3.7M |
| Encoder GFLOPs | 1.4 | 17.4 | 15.0 |
| Decoder GFLOPs | 2.5 | 25.3 | 37.1 |
| **Training Details** | | | |
| Minibatch size | 128 | 1024 | 800 |
| Batch per device | 128 | 512 | 400 |
| Iterations | 400k | 400k | 400k |
| Dropout probability | 20% | 20% | 20% |
| Optimizer | RAdam | RAdam | RAdam |
| Learning rate | 0.0001 | 0.0001 | 0.0001 |
| EMA rate | 0.9999 | 0.9999 | 0.9999 |
| Gradient clip value | 200 | 200 | 200 |
| Number of GPUs | 1 | 2 | 2 |
| GPU types | A100 | H100 | H100 |

Table 3: Model configurations and training details for CoVAE for the different datasets.

## B.4 INITIAL TUNING PHASE

To find a suitable set of hyperparameters for CoVAE, we use the architecture with 35.8M parameters with batch size 128 and $3\times$ compression rate on CIFAR-10. In early experiments we used $\lambda(t) = 1/t$ and $\beta(t) = t$, but we found the reconstruction loss to become unstable for small value of t. We therefore changed $\beta(t) = t^2$ to allow for more faithful reconstruction at early time steps without the need to lower $\sigma_{\min}$ too much. We further did a grid search with the following hyperparameters, with $s_0 = 2$ and $s_1 = 256$:

- $\sigma_{\min} = [0.01, 0.05, 0.1, 0.2]$;

- $\sigma_{\max} = [1, 1.5, 2, 3, 4, 5]$;

- $\rho = [3, 5, 7]$.

These experiments were run with dropout probability 20%. Afterwards, we experimented with dropout rates $[0\%, 10\%, 20\%, 30\%]$ for the best model with $\sigma_{\min} = 0.05$, $\sigma_{\max} = 3$ and $\rho = 7$, and found 20% dropout rate to work best. We use these hyperparameters also on MNIST and CelebA 64 without further tuning. For the $\beta$-VAE baseline we tuned $\beta$ with the same settings, and searched with the values $\beta = [0.05, 0.1, 0.5, 1.5, 2]$, and found $\beta = 0.1$ to work best (while $\beta = 0.5$ worked best for MNIST).

## B.5 MULTISTEP SAMPLING

To find the optimal time step for multi-step sampling, we first try all the available steps after training, and select the one that gives the best 2-steps FID. We then repeat the procedure for 3 and 4 steps, keeping fixed the time steps found at the previous iteration. While this might not be optimal for more than 2 sampling iterations, we believe it can already provide a good enough heuristic for finding good multi-step sampling times. For MNIST, we use $t = 0.8538$ (idx=162) for 2-steps. For CIFAR-10 and CelebA 64 we test 2, 3 and 4 steps, corresponding to 3, 5 and 7 NFEs, and report the FID results and corresponding time steps in table 4. While increasing the sampling steps results in lower FID, the improvements decrease as we use more sampling iterations. Perhaps counterintuitively, in some cases adding an extra sampling iteration achieves improved results when re-adding noise at a bigger time step than the iteration before.

| Model, Data | Time steps | Indexes ($\in [1, 257]$) | FID |
|---|---|---|---|
| CoVAE, CIFAR-10 | [1.412, 0.6745, 0.7266] | [198, 146, 151] | [14.06, 13.35, 13.01] |
| CoVAE w/ $\mathcal{L}_{\text{adv}}$, CIFAR-10 | [2.4343, 2.3447, 2.0397] | [240, 237, 226] | [9.82 , 9.19, 8.83] |
| CoVAE w/ $\mathcal{L}_{\text{adv}}$, CelebA 64 | [1.9376, 2.1193, 2.0659] | [222, 229, 227] | [7.15, 6.98, 6.82] |

Table 4: Multistep FID results and corresponding time steps and time indexes. The results are reported as [2-steps, 3-steps, 4-steps] in the corresponding lists.

### B.6 DATA PROCESSING

For all the datasets we rescale the values to be in the range [-1,1]. For CIFAR-10 and CelebA 64 we also apply random horizontal flip with $50\%$ probability. For CelebA 64, we first take the center crop of size $148 \times 148$ and then resize to $64 \times 64$ as done in Xiao et al. (2021).

## C ADDITIONAL RESULTS

### C.1 ABLATION ON CIFAR-10

As a base model, we select an architecture with 32.8M parameters, latent size 1024, and batch size 128. By varying these factors in isolation, we can see the effect of each on the generative performance, measured in FID. The remaining hyperparameters are the same as outlined in Appendix B. We report the results of the ablation in figure 6, where we note how CoVAE greatly benefits from bigger latent size, while also improving as batch size and model parameters increase. In the bottom-right plot, we compare the effect of different losses. The first two bars labeled "L2" and "ph" correspond to training the model without boundary condition from equation 13, and with L2 and pseudo-huber loss respectively. The others are a combination of the two, where for example "L2+ph" means that the average decoder network $\hat{x}_{\theta}$ is trained with L2 loss and the ovreall decoder $\mathcal{D}_{\theta}$ is trained with pseudo-huber loss. Note that using only the pseudo-huber loss like in the second column usually leads to instabilities, and the model diverges during training.

### C.2 BROADER COMPARISON ON CIFAR-10

We report in table 5 the FID results on CIFAR-10 from a broad selection of generative models, as CIFAR-10 is a commonly used baseline. While many direct generation methods achieve superior quality cmpared to CoVAE, CoVAE is among the best few-step methods to work in latent space, and without the need for a two-stage training procedure. The NFEs for latent models accounts for function evaluations needed for the prior plus the decoder.

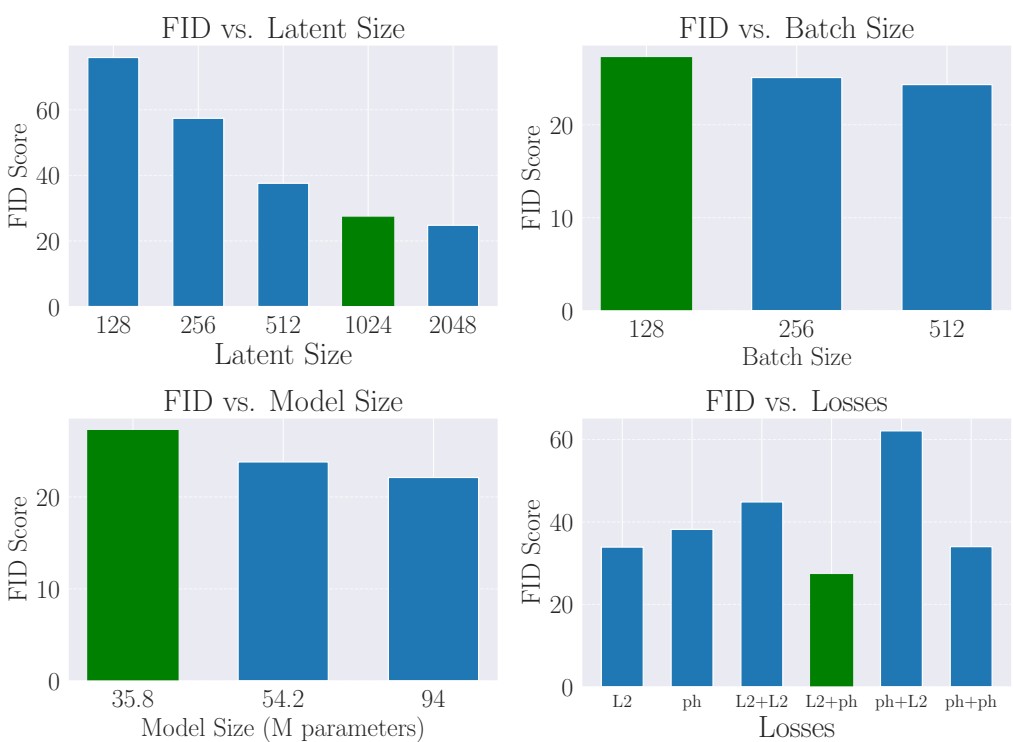

Figure 6: Visualization of the 1-step FID performance (lower is better) for CoVAE with varying hyperparameters. The green bar corresponds to the same run.

| METHOD | NFE ($\downarrow$) | FID ($\downarrow$) |
|---|---|---|
| **VAE-Based** | | |
| NVAE (Vahdat & Kautz, 2020) | 1 | 23.49 |
| DC-VAE (Parmar et al., 2021) | 1 | 17.9 |
| VAEBM (Xiao et al., 2021) | 17 | 12.19 |
| **Diffusion-based** | | |
| DDIM (Song et al., 2021a) | 10 | 8.23 |
| | 20 | 6.84 |
| | 50 | 4.67 |
| DPM-Solver (Lu et al., 2022) | 10 | 4.70 |
| DPM-Solver++ (Lu et al., 2025) | 10 | 2.91 |
| DPM-Solver-v3 (Zheng et al., 2023) | 10 | 2.51 |
| Score SDE (Song et al., 2021b) | 2000 | 2.20 |
| DDPM (Ho et al., 2020) | 1000 | 3.17 |
| Flow Matching (Lipman et al., 2023) | 142 | 6.35 |
| EDM (Karras et al., 2022) | 35 | 2.04 |
| **Diffusion + VAE** | | |
| D2C (Sinha et al., 2021) | 11 | 17.71 |
| | 51 | 10.11 |
| | 101 | 10.15 |
| DiffuseVAE (Pandey et al., 2022) | 1001 | 2.62 |
| LSGM (Vahdat et al., 2021) | 147 | 2.10 |
| **Diffusion Distillation** | | |
| PD (Salimans & Ho, 2022) | 1 | 8.34 |
| | 2 | 5.58 |
| TRACT (Berthelot et al., 2023) | 1 | 3.78 |
| | 2 | 3.32 |
| CD (LPIPS) (Song et al., 2023) | 1 | 3.55 |
| | 2 | 2.93 |
| sCD (Lu & Song, 2025) | 1 | 3.66 |
| | 2 | 2.52 |
| CTM (Kim et al., 2024) | 2 | 1.87 |
| **Direct Generation** | | |
| Glow (Kingma & Dhariwal, 2018) | 1 | 48.9 |
| Residual Flow (Chen et al., 2019) | 1 | 46.4 |
| BigGAN (Brock et al., 2019) | 1 | 14.7 |
| StyleGAN2 (Karras et al., 2020b) | 1 | 9.26 |
| StyleGAN2-ADA (Karras et al., 2020a) | 1 | 2.92 |
| CT (LPIPS) (Song et al., 2023) | 1 | 8.70 |
| | 2 | 5.83 |
| iCT (Song & Dhariwal, 2024) | 1 | 2.83 |
| | 2 | 2.46 |
| sCT (Lu & Song, 2025) | 1 | 2.85 |
| | 2 | 2.06 |
| IMM (Zhou et al., 2025) | 1 | 3.20 |
| | 2 | 1.98 |
| MeanFLow (Geng et al., 2025a) | 1 | 2.92 |
| **Ours** | | |
| CoVAE | 1 | 17.21 |
| | 3 | 14.06 |
| CoVAE w/ $\mathcal{L}_{\mathrm{adv}}$ | 1 | 11.69 |
| | 3 | 9.82 |

Table 5: Comparison of FID performance with a broad selection of generative models.

# D QUALITATIVE RESULTS

## D.1 LATENT SPACE VISUALIZATION FOR MNIST

The CoVAE model trained on MNIST has a latent size spatially organized as a $7 \times 7$ grid. This allows us to visualize the latents as grayscale images and get a visual understanding of the learned latent dynamics. In figure 7 we show the learned mean and standard deviation for some of the training images, while varying the time step of the embedding. We further pair each image with a latent noise mask, and show the corresponding sample from the encoded distribution. At small time steps, the encoded means resembles a downscaled version of the input images, and the standard deviations are generally small, resulting in samples indistinguishable from the means. As the time step increases, the mean values get closer to zero and the standard deviations closer to one, resulting in posterior samples which are almost identical to the noise mask. Note that it is not necessary for each encoded distribution at time $t = \sigma_{\max}$ to perfectly match the prior, i.e. isotropic Gaussian, but as in VAEs, we need the aggregate posterior to recover the prior.

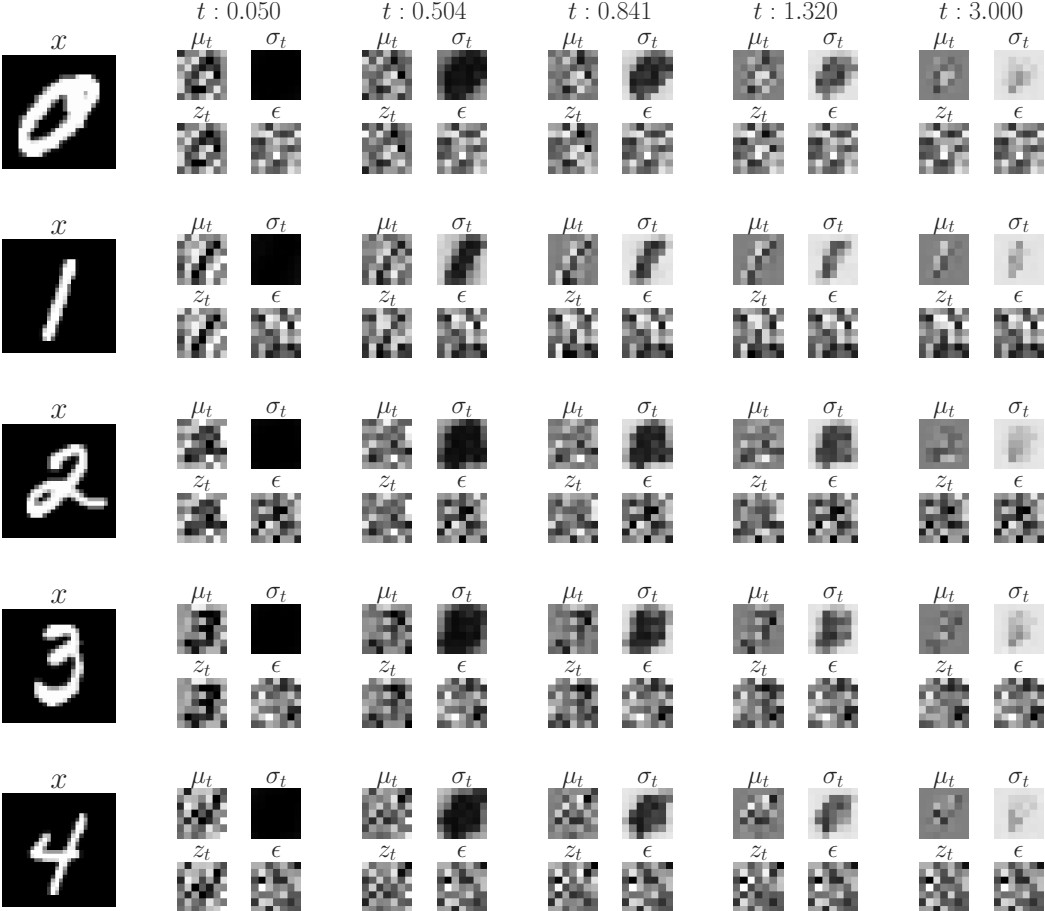

Figure 7: Visualization of latent space learned by CoVAE for different time steps.

## D.2 GENERATED SAMPLES

We report here additional samples from our models, on MNIST in figure 8, CIFAR-10 in figures 9 and 10, and on CelebA 64 in figures 11. Zoom in for best results.

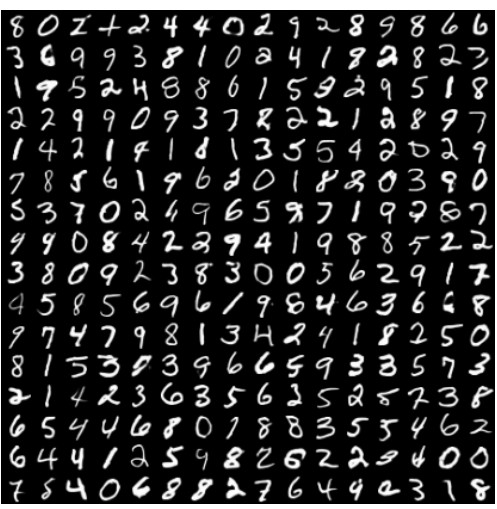 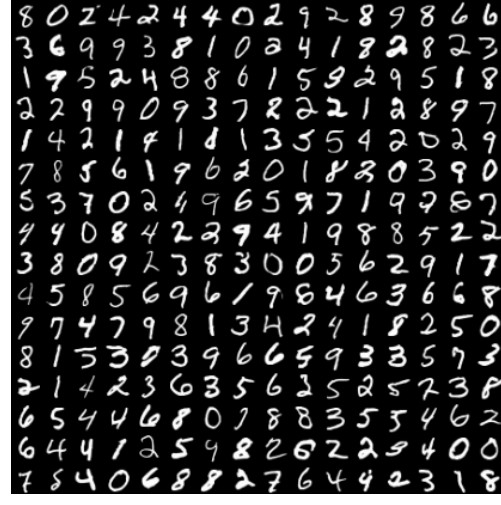

Figure 8: 1-step (FID=5.62, left) and 2-step (FID=3.83, right) generation from CoVAE on MNIST.

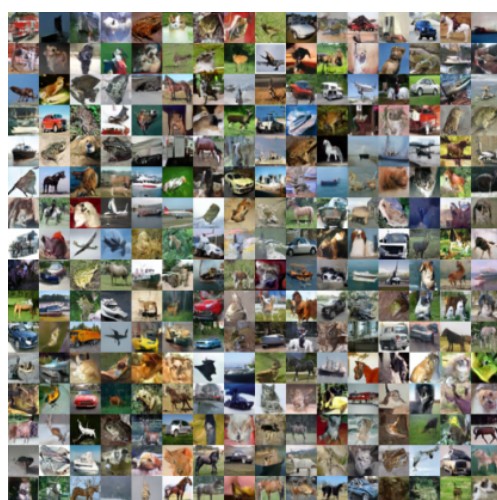 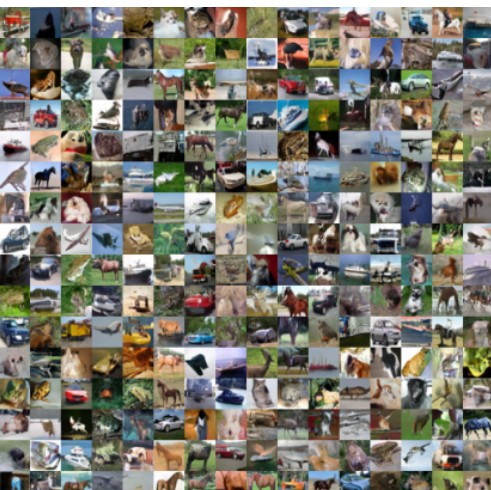

Figure 9: 1-step (FID=17.21, left) and 2-step (FID=14.06, right) generation from CoVAE on CIFAR-10.

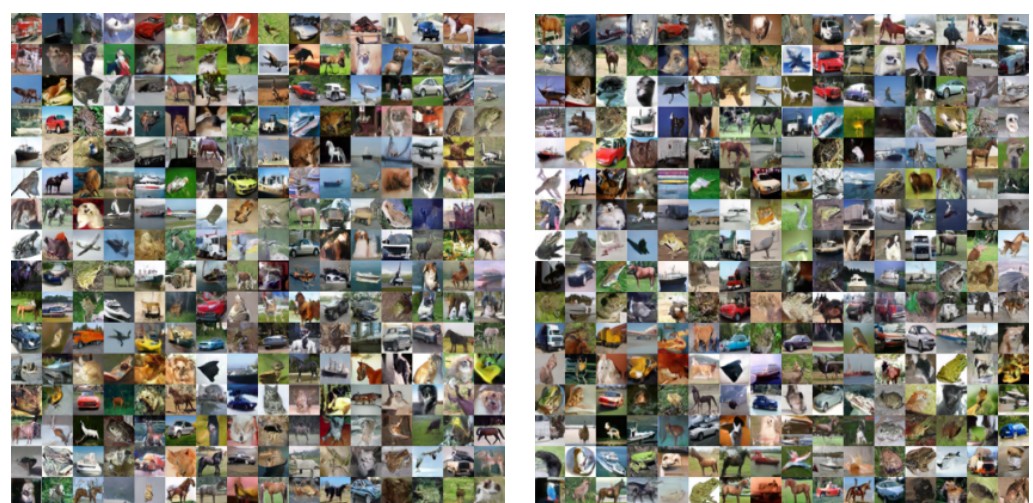

Figure 10: 1-step (FID=11.69, left) and 2-step (FID=9.82, right) generation from CoVAE w/ $\mathcal{L}_{adv}$ on CIFAR-10.

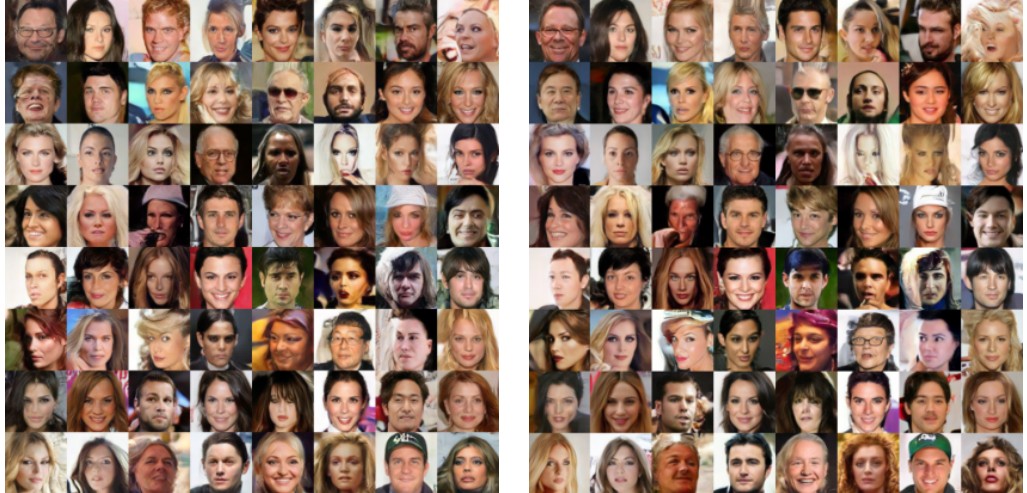

Figure 11: 1-step (FID=8.27, left) and 2-step (FID=7.15, right) generation from CoVAE w/ $\mathcal{L}_{adv}$ on CelebA 64.

### D.3 Latent interpolation on CelebA 64

In this section, we analyze the effect of image latent space interpolation at different time steps, where a scalar value $\alpha$ is used to interpolate between latent vectors $\boldsymbol{z}_0$ and $\boldsymbol{z}_1$ sampled from the embeddings of two different images $\boldsymbol{x}_0$ and $\boldsymbol{x}_1$ with the same random direction $\boldsymbol{\epsilon}$. We show the reconstructions from the interpolations for different $\alpha$ and different time steps in figures 12 and 13. For small time steps, while the reconstructions without interpolation are almost perfect, we can notice overlapping of the two original images for intermediate values of $\alpha$, especially for images with very distinct features. As $t$ increases, the reconstructions get further away from the original input, but the interpolations transition smoothly between the two images, indicating better latent space disentanglement.

### D.4 Latent manipulation on CelebA 64

Similarly to what done in other VAE works such as Parmar et al. (2021); Pandey et al. (2022), we show some results from latent space manipulation using CelebA 64, as it has 40 annotated binary attributes per image. To add or remove one of such attributes, we first compute an estimate of the latent direction $\boldsymbol{z}_a$ of that attribute by encoding $N$ images with the attribute and $N$ without, sampling from the respective latent spaces, obtaining $\boldsymbol{z}_p$ for positive latent vectors and $\boldsymbol{z}_n$ negatives, and then subtracting the respective means as:

$$\boldsymbol{z}_a = \frac{1}{N} \sum (\boldsymbol{z}_p) - \frac{1}{N} \sum (\boldsymbol{z}_n), \qquad (19)$$

where we set $N = 100$. The modified latent of an encoded image that does not have the selected attribute is computed with the following:

$$\boldsymbol{z}' = \boldsymbol{z} + \psi \boldsymbol{z}_a, \qquad (20)$$

where $\psi$ is a scalar that regulates the strength of the update. Similarly, to remove an attribute one can simply subtract $\boldsymbol{z}_a$ instead. As CoVAE can obtain latent representations at different time steps, we show the effect of latent manipulation at different time steps and for different manipuation strength in figures 14 and 15, obtained with CoVAE /w $\mathcal{L}_{\mathrm{adv}}$. For the modifications at small time steps to become visible, a bigger $\psi$ is needed, which also seems to introduce some artifacts, but can obtain a faithful reconstruction to the original image. For bigger time steps, the modifications tend to be more visible already with small $\psi$, and increasing $\psi$ has less visible artifacts, but the reconstructed image is further away from the original input.

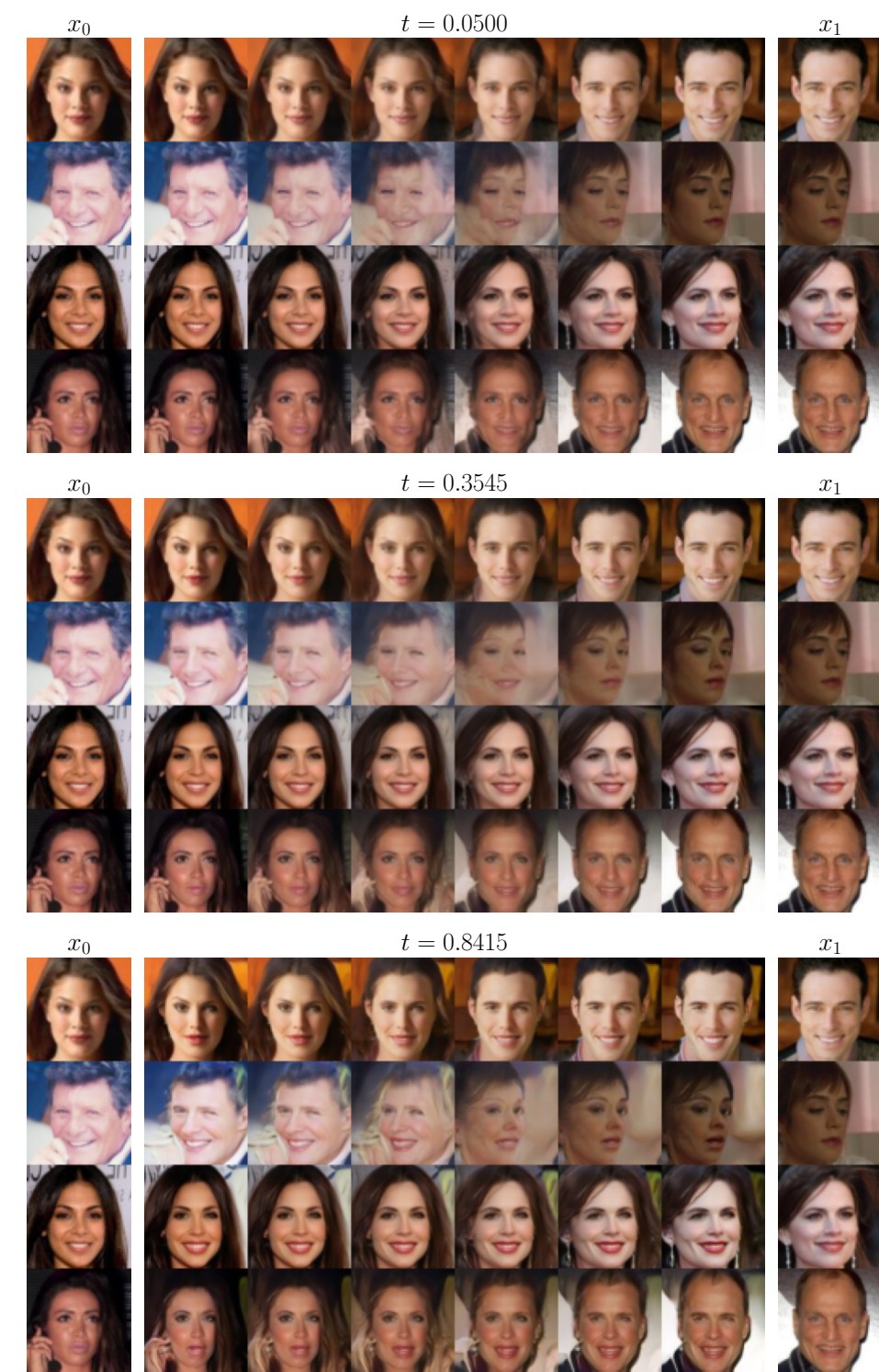

Figure 12: The figure shows the reconstruction from latent space interpolation between two data points displayed on the right and left hand side columns. The interpolations are obtained with mixing factor $\alpha \in [0, 0.2, 0.4, 0.6, 0.8, 1]$ from left to right in the central grid. The embeddings are obtined with time step $t$ displayed on top.

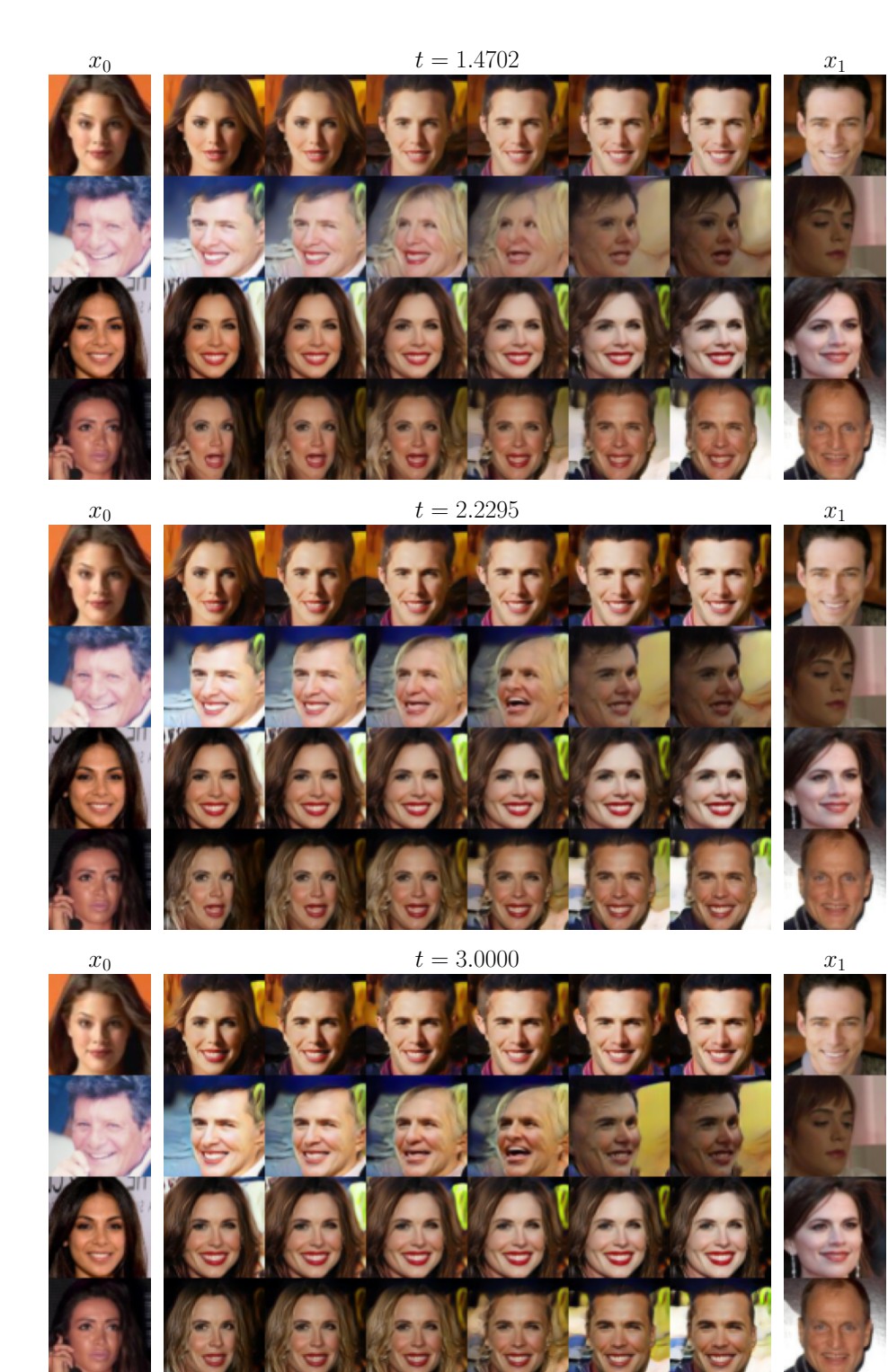

Figure 13: Continuation of figure 12 for higher values of t.

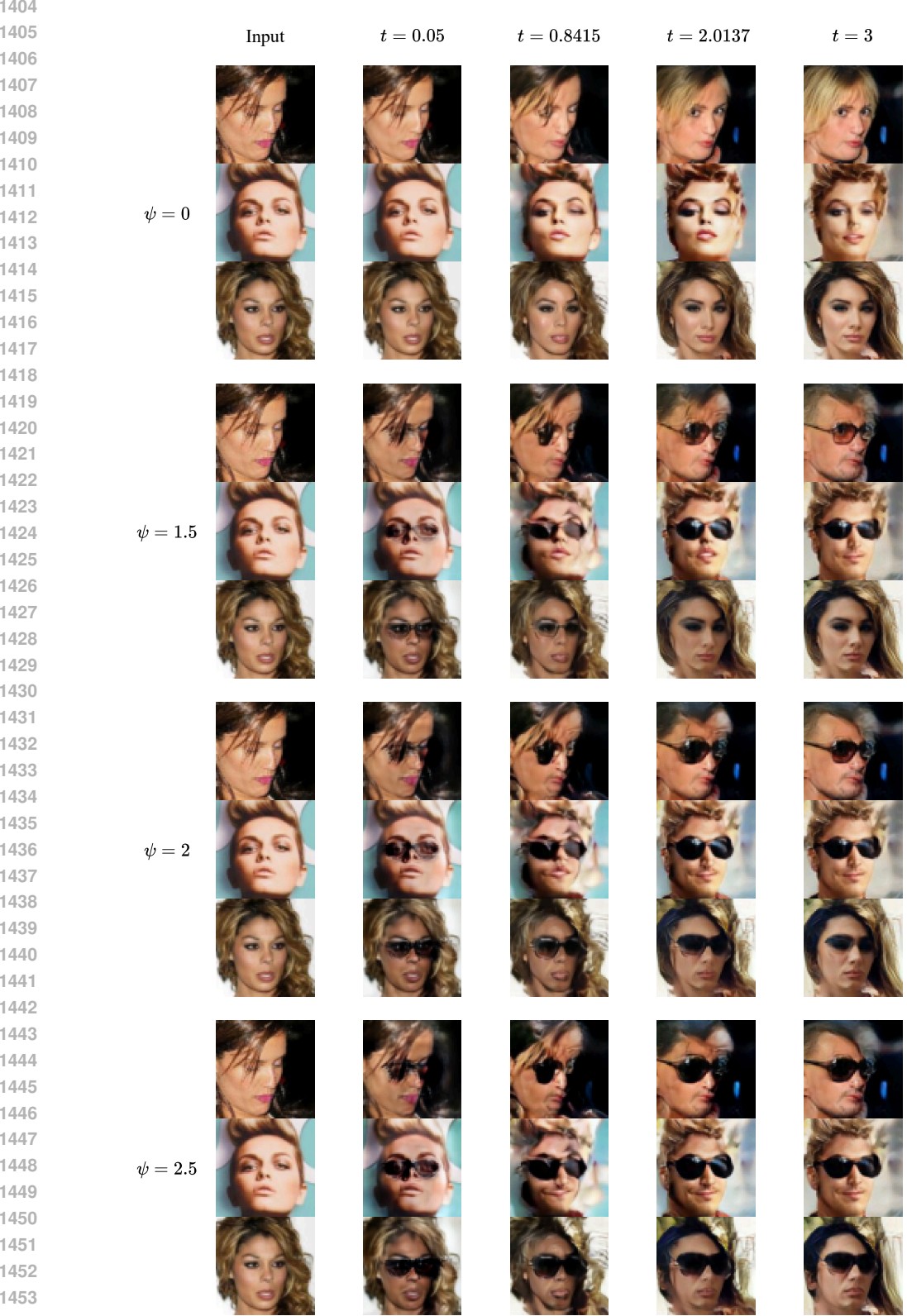

Figure 14: Latent space manipulation experiments adding the latent direction for the attribute "Eyeglasses". The first row with $\psi = 0$ corresponds to no manipulation, and is used to show the reference reconstructed embedding.

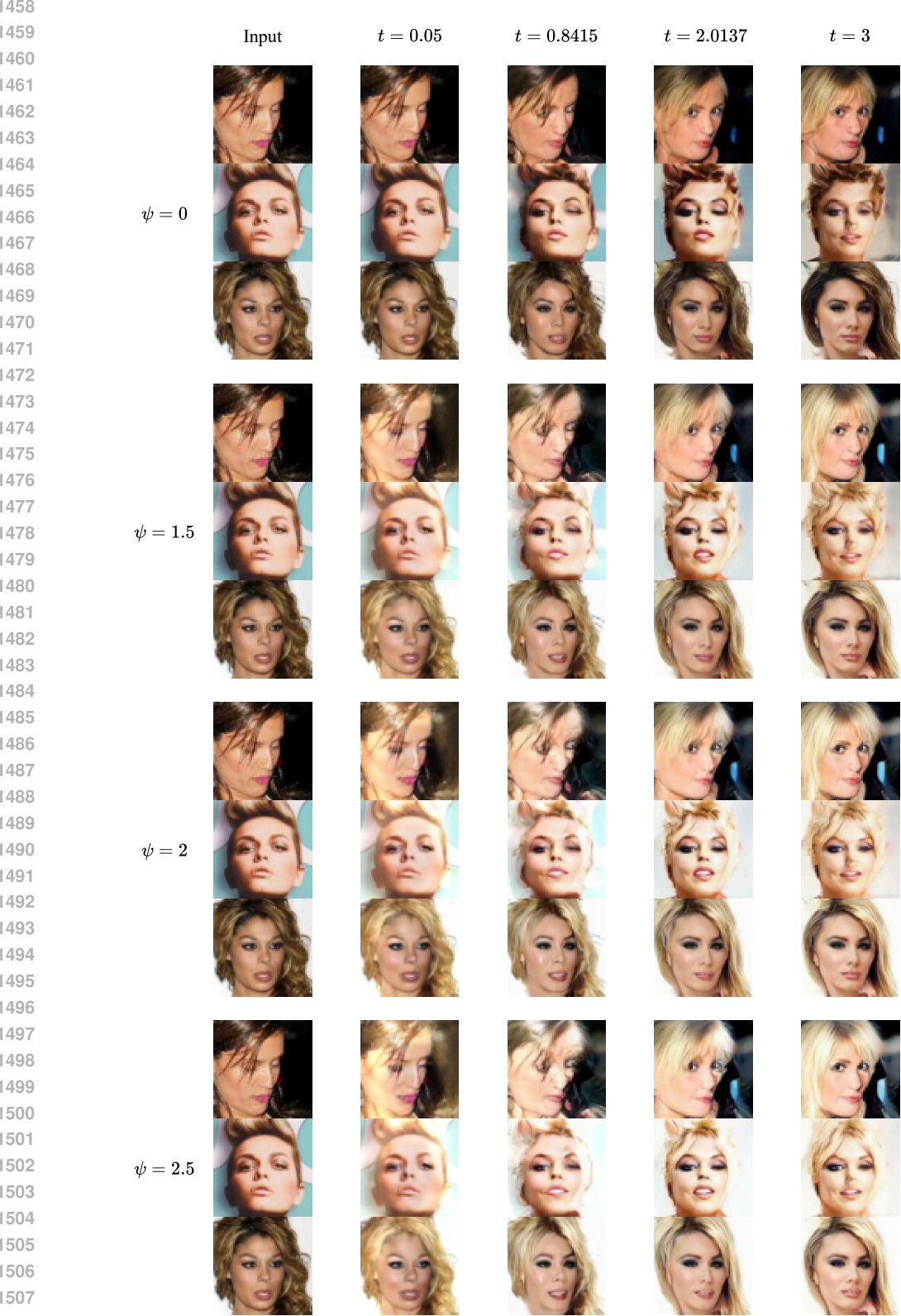

Figure 15: Latent space manipulation experiments adding the latent direction for the attribute "Blonde". The first row with $\psi = 0$ corresponds to no manipulation, and is used to show the reference reconstructed embedding.

# E  ALTERNATIVE FORMULATIONS FOR COVAE

In this section we discuss alternative formulations for CoVAE. We first show how CoVAE can be simplified to reuse components from CMs while being more efficient to train. We then show how CoVAE can be used with discrete data. While these formulations did not match CoVAE's performance in simple benchmarks, we believe they can be interesting twists on the main framework, and we leave the scaling and improvements of such variants to future work.

## E.1  SIMPLIFIED COVAE

In its original formulation, CoVAE learns a time-dependent embedding that mimicks the forward processes commonly used in DMs. This mechanism has the downside of heavily relying on the choice of weighting functions $\lambda(t)$ and $\beta(t)$, as well as a careful tuning of $\sigma_{\min}$ and $\sigma_{\max}$. Furthermore, there is no analytic formulation of the SNR at different time steps, which is an important quantity used to design several components for training and sampling with DMs (see for example Kingma et al. (2021)). In this section, we show that the CoVAE formulation can be modified to use standard forward processes commonly used in DMs, while reducing the need for several empirical components. In particular, we can formulate the time-dependent encoder as a combination of an encoder without time dependence and a forward process in latent space:

$$\boldsymbol{z}_t = a_t \boldsymbol{\mathcal{E}}_{\phi}(\boldsymbol{x}) + b_t \boldsymbol{\epsilon}, \quad \boldsymbol{\epsilon} \sim \mathcal{N}(\mathbf{0}, \boldsymbol{I}) \tag{21}$$

where $a_t$ and $b_t$ are time dependent scalar functions. A valid choice is, for example, $a_t = 1$ and $b_t = t$ like in the Variance Exploding case. With this formulation, the latent space transitions to a Gaussian distribution without requiring the $\lambda(t)$ and $\beta(t)$ regularization terms, and we can reuse most of the components used for training DMs and CMs. However, we need to add a regularization for the encoder, to avoid it learning arbitrarily large embeddings. To this end, we add a latent regularization of the form $\gamma \|\boldsymbol{\mathcal{E}}_{\phi}(\boldsymbol{x})\|^2$ to the reconstruction loss in equation 14, where $\gamma$ is a scalar hyperparameter ($\gamma = 0.001$ in our experiments). Such a regularization is equivalent to imposing a KL loss with fixed variance. Alternatively, to further simplify the model and removing the need for extra hyperparameters, we simply normalize the output of the encoder with LayerNorm (Ba et al., 2016) followed by tanh activation. The training procedure for this simplified CoVAE (s-CoVAE) model is the same as for CoVAE, but is more efficient as the encoder network needs to be used only once for each training example (instead of twice), reusing the encoded representation and applying the latent forward kernel at both times $t_i$ and $t_{i-1}$. We train s-CoVAE by reusing the same strategy as iCT from (Song & Dhariwal, 2024), with $\sigma_{\min} = 0.002$, $\sigma_{\max} = 80$, $s_0 = 10$, $s_1 = 1280$, and the Variance Exploding kernel. For the encoder, we do not use adaptive GroupNorm as the model is not time conditioned. As now we know how the forward process progressively adds noise to the data, we can reuse the scaling factors $c_{\text{skip}}(t)$ and $c_{\text{out}}(t)$ from iCT in the boundary conditions in equation 13. However, as the learned average decoder is a different quantity, we need a custom value for $c_{\text{skip}}(t)$. In the following section, we will show that setting $c_{\text{skip}}(t) = 1$ matches the variance of $c_{\text{skip}}(t)\boldsymbol{x}_t$ in CMs, while we keep $c_{\text{out}}(t)$ and $c_{\text{in}}(t)$ as in iCT. Similarly to CoVAE, we add an average decoder loss scaling factor $\lambda_d(t) = \sigma_{\text{data}}^2/(t^2 + \sigma_{\text{data}}^2)$ (corresponding to $c_{\text{skip}}(t)$ from iCT). We test this formulation on CIFAR-10 using the same settings from section C.1 (32.8M parameters, latent size 1024, and batch size 128). The model with $\gamma$ regularization achieves 1-step FID of 40.42, while the method with normalization performs slightly better, with 38.18 1-step FID. In this experiment, CoVAE performs better than s-CoVAE (27.21 FID), likely thanks to the learned latent forward dynamics, but we believe that s-CoVAE, if scaled properly, can be a viable alternative as it is faster to train and requires less hyperparameters.

### E.1.1  DERIVATION OF THE AVERAGE DENOISER IN VE DIFFUSION

We consider the Variance Exploding (VE) forward process:

$$\boldsymbol{\mathcal{F}}(\boldsymbol{x}, t) = \boldsymbol{x} + t\boldsymbol{\epsilon}, \quad \boldsymbol{\epsilon} \sim \mathcal{N}(\mathbf{0}, \boldsymbol{I}) \tag{22}$$

Assume the data distribution is $\boldsymbol{x} \sim \mathcal{N}(\mathbf{0}, \sigma_{\text{data}}^2 \boldsymbol{I})$. We want to compute the posterior mean:

$$\hat{x}_t = \mathbb{E}[\boldsymbol{x} \mid \boldsymbol{x}_t] \tag{23}$$

This is a Gaussian denoising problem: we observe $\boldsymbol{x}_t$ which is a noisy version of $\boldsymbol{x}$. Since both $\boldsymbol{x}$ and the noise are Gaussian, the posterior $p(\boldsymbol{x} \mid \boldsymbol{x}_t)$ is also Gaussian. Let:

$$p(\boldsymbol{x}) = \mathcal{N}(\boldsymbol{x}; 0, \sigma_{\text{data}}^2 \boldsymbol{I})$$

$$p(\boldsymbol{x}_{\rangle} \mid \boldsymbol{x}) = \mathcal{N}(\boldsymbol{x}_t; \boldsymbol{x}, t^2 \boldsymbol{I})$$

Then by Bayes' rule:

$$p(\boldsymbol{x} \mid \boldsymbol{x}_t) \propto p(\boldsymbol{x}_t \mid \boldsymbol{x}) p(\boldsymbol{x}) \tag{24}$$

This is a standard case of Gaussian conjugate priors. The posterior mean is given by:

$$\hat{\boldsymbol{x}}_t = \left( \frac{1}{\sigma_{\text{data}}^2} + \frac{1}{t^2} \right)^{-1} \cdot \left( \frac{\boldsymbol{x}_t}{t^2} \right) \tag{25}$$

Simplifying:

$$\hat{\boldsymbol{x}}_t = \frac{\sigma_{\text{data}}^2}{\sigma_{\text{data}}^2 + t^2} \boldsymbol{x}_t \tag{26}$$

### E.1.2 Variance and Standard Deviation

The variance of the posterior mean across samples $\boldsymbol{x}_t$ is:

$$\text{Var}(\hat{\boldsymbol{x}}_t) = \left( \frac{\sigma_{\text{data}}^2}{\sigma_{\text{data}}^2 + t^2} \right)^2 \cdot \text{Var}(\boldsymbol{x}_t) \tag{27}$$

Since:

$$\text{Var}(\boldsymbol{x}_t) = \text{Var}(\boldsymbol{x} + t\boldsymbol{\epsilon}) = \sigma_{\text{data}}^2 + t^2 \tag{28}$$

We have:

$$\text{Var}(\hat{\boldsymbol{x}}_t) = \frac{\sigma_{\text{data}}^4}{\sigma_{\text{data}}^2 + t^2} \tag{29}$$

And therefore, the standard deviation is:

$$\text{Std}(\hat{\boldsymbol{x}}_t) = \frac{\sigma_{\text{data}}^2}{\sqrt{\sigma_{\text{data}}^2 + t^2}} \tag{30}$$

### E.1.3 Boundary conditions with average denoiser

The boundary conditions commonly used in CMs are of the form:

$$f_\theta(\boldsymbol{x}_t, t) = c_{\text{skip}}(t)\boldsymbol{x}_t + c_{\text{out}}(t)\boldsymbol{F_\theta}(\boldsymbol{x}_t, t) \tag{31}$$

with $c_{\text{skip}}(\sigma_{\min}) = 1$ and $c_{\text{out}}(\sigma_{\min}) = 0$. For the VE case, $c_{\text{skip}}(t)$ is defined as:

$$c_{\text{skip}}(t) = \frac{\sigma_{\text{data}}^2}{t^2 + \sigma_{\text{data}}^2} \tag{32}$$

The variable $c_{\text{skip}}(t)$ is used to multiply $\boldsymbol{x}_t$ which has a standard deviation of $\sqrt{t^2 + \sigma_{\text{data}}^2}$, so that $\text{STD}(c_{\text{skip}}(t)\boldsymbol{x}_t) = \frac{\sigma_{\text{data}}^2}{\sqrt{\sigma_{\text{data}}^2 + \sigma^2(t)}}$. As we can see, this is already equivalent to the standard deviation of the average denoiser $\hat{\boldsymbol{x}}_t$, so the new boundary condition is simply:

$$f_\theta(\boldsymbol{x}_t, t) = \hat{\boldsymbol{x}}_t + c_{\text{out}}(t)\boldsymbol{F_\theta}(\boldsymbol{x}_t, t), \tag{33}$$

where in practice we approximate the average decoder/denoiser with a neural network.

### E.2 CoVAE with discrete data

Differently than DMs and CMs that work in ambient space, CoVAE learns a data-dependent mapping to the latent space, which allows us to work with both discrete and continuous data. While using CoVAE with discrete data is not the focus of this work, we report here results from a prof of concept on binary MNIST, where instead of the L2 and pseudo-huber loss, we use the binary cross-entropy loss. In Figure 16 we show 1-step samples from CoVAE, trained with the same hyperparameters used in 4, apart from $\sigma_{\min}$ that was raised to 0.5, achieving 1-step sampling FID of 0.58 and reconstruction FID of 0.17. While from this simple experiment it is not easy to tell how scalable the setting can be to complex discrete data such as text and biological data, we believe it can be an interesting direction for future research.

Figure 16: 1-step samples from binary MNIST, FID=0.58.

