# OpenReview forum: "CoVAE: Consistency Training of Variational Autoencoders"
_ICLR.cc/2026/Conference — Submitted to ICLR 2026_

### Official Review · Reviewer_DFFZ · 2025-10-27

**Soundness:** 3
**Presentation:** 3
**Contribution:** 2
**Rating:** 2
**Confidence:** 4

**Summary:**

The paper introduices CoVAE which builds upon the VAE architecture but modifies both the encoding and decoding processes to ensure temporal consistency across latent representations along the learning phase.
•	The encoder learns progressively noisier latent representations, controlled by a time-dependent noise schedule β(t). At low noise levels (small t), the model captures detailed structure; as t increases, the latent codes gradually transition to Gaussian noise, providing a continuous interpolation between structured and random representations.
•	The decoder is trained using a consistency loss that enforces agreement between predictions at adjacent time steps. This mechanism enables the model to learn denoising dynamics similar to those in diffusion models, but without requiring iterative multi-step sampling.

**Strengths:**

•	Single-stage training (faster, simpler)
•	Fast sampling without leaning an adaptive priori or learning the posterior distribution of the embeddings.
•	Disentangled latent space for image manipulation

**Weaknesses:**

The model cannot easily compute a tight Evidence Lower Bound (ELBO), which complicates likelihood-based evaluation and comparison with classical VAEs.
Its performance depends on several empirically tuned hyperparameters.
While it closes part of the gap between VAEs and diffusion models, it still lags behind the best direct diffusion approaches in sample fidelity and comparisons with the latent diffusion models are lacking.
Other Latent SDE approached should also have been considered for comparison.

Overall the idea looks interesting to have a "diffusion like" model to better learn the latent prior without the diffusion forward-backward equations to solve but comparisons with this literature is missing.

There is a typo in the pseudo code Algorithm 1 for the cm likelihood: , should be -.

**Questions:**

Is the multistep CoVAE remaining in interesting areas of the latent space when moving around? In particular, the experiments have been performed with a lot of data. Is this method "scalable" for low amount of data?

Is the use of patch-based adversarial loss interesting for the β-VAE itself. If yes, can you show the increase of performance of your combined approach with CoVAR vs the combination with the β-VAE?

---

> ### Author Response · Authors · 2025-11-21
>
> We thank the reviewer for the thoughtful questions.
>
> ## Weakness
> > The model cannot easily compute a tight Evidence Lower Bound (ELBO), which complicates likelihood-based evaluation and comparison with classical VAEs. Its performance depends on several empirically tuned hyperparameters. While it closes part of the gap between VAEs and diffusion models, it still lags behind the best direct diffusion approaches in sample fidelity and comparisons with the latent diffusion models are lacking. Other Latent SDE approached should also have been considered for comparison.
> Overall the idea looks interesting to have a "diffusion like" model to better learn the latent prior without the diffusion forward-backward equations to solve but comparisons with this literature is missing.
>
> Regarding the comparison with broader classes of generative models: we defer the reader to Appendix A and Table 5, where we provide an extended discussion of related methods, including latent-diffusion-based approaches. We emphasize, however, that CoVAE is conceptually and architecturally much closer to a VAE than to a latent diffusion model: it uses only an encoder–decoder architecture, performs one-step (or few-step) sampling, and is trained end-to-end without relying on a learned prior, a separate score model, or complex posterior distributions.
>
> Importantly, in standard latent diffusion pipelines, the VAE component is trained separately and in advance, before learning the diffusion model in the latent space. In contrast, CoVAE integrates representation learning and generative modeling into a single unified objective. This also means that CoVAE can, in principle, serve as a drop-in replacement for the VAE stage in latent diffusion systems, should one wish to use a consistency-based autoencoder instead of a conventional one. For these reasons, we frame CoVAE primarily as an advancement within the family of generative autoencoders rather than as a direct competitor to multi-stage latent diffusion pipelines.
>
> ## Typo:
> > There is a typo in the pseudo code Algorithm 1 for the cm likelihood: , should be -.
>
> The CoVAE loss is a sum of reconstruction term and KL regularization, so we believe it is correct as is in the pseudocode.
>
> ## Question 1
> > Is the multistep CoVAE remaining in interesting areas of the latent space when moving around? In particular, the experiments have been performed with a lot of data. Is this method "scalable" for low amount of data?
>
> We have not explicitly evaluated CoVAE in the low-data regime, but we expect its behavior to be similar to that of consistency and diffusion-based models. On one hand, the consistency loss provides additional regularization at large t values, which may help stabilize training under limited data. On the other hand, the model still relies on learning a meaningful time-dependent encoding distribution, which, like VAEs and consistency models, will naturally benefit from sufficient data. Investigating few-shot and low-data performance is an interesting avenue for future work.
>
> Concerning latent-space behavior (e.g., whether multi-step CoVAE stays within meaningful regions of the latent manifold), this is an important question. Although CoVAE produces structured latents similar to other autoencoder-based models, a detailed evaluation of latent geometry and downstream use remains beyond the scope of this work and is left for future investigation.
>
> ## Question 2
> > Is the use of patch-based adversarial loss interesting for the β-VAE itself. If yes, can you show the increase of performance of your combined approach with CoVAR vs the combination with the β-VAE?
>
> We trained our best-performing β-VAE using the same adversarial loss configuration adopted for CoVAE. However, training diverged, as the discriminator rapidly learned to perfectly distinguish real from generated samples, and the β-VAE could not produce sufficiently realistic outputs. Training a β-VAE with small β like in the latent diffusion setup would lead to great reconstruction quality, but no generation capability, unless combined with a powerful generative prior.

---

### Official Review · Reviewer_ovEJ · 2025-10-30

**Soundness:** 3
**Presentation:** 2
**Contribution:** 2
**Rating:** 4
**Confidence:** 4

**Summary:**

The paper introduces consistency training for Variational AutoEncoders (CoVAEs). In contrast with two-stage approaches based on generative models over dimensionality reduction, CoVAE is a one-stage approach that learns a sequence of latent representations with a time dependent parameter scaling regularization and trained using consistency loss that is related to standard the VAE. Experiments on MNIST, CIFAR-10 and CelebA comparing performance with baseline models to illustrate the quality of images generated.

**Strengths:**

Strengths of the paper include:
- Concise, clear mathematical introduction of VAEs, Diffusion models, Consistency models, and the proposed CoVAE approach.
- Detailed experiments including multiple datasets and multiple baseline models
- Detailed and fair discussion of related work
- Clear statements of limitations of the current work that identify important problems to address in future research

**Weaknesses:**

The main weaknesses of the paper are twofold;
- First, the datasets are limited to MNIST, CIFAR10, and CelebA
- Second, the models compared with are valuable but not state-of-the-art in terms of performance


Minor issues:
- Small typo in Figure 1 caption: Consistenct
- Line 229 "Form small time steps"
- Figure 2 is confusing. I suggest you explain what the objects in the future are one by one, starting from the left. E.g. is "In Diffusion and Consistency" about the first picture or the first two? Epsilon_psi is in in the figure but is in the caption. "in this case we use a dashed line" This is confusing.
- Figure 3, the caption never mentions t.
- "Form small time steps, the samples from each class are embedded in well separated areas, while they gradually become more random
as time increases." This is unclear. Is t representing size of time steps or time? (Or are they confounded.)

**Questions:**

Please see weaknesses. Specifically, it seems important to better understand the implications of the current approach for the state-of-the-art in image generation?

---

> ### Author Response · Authors · 2025-11-21
>
> We thank the reviewer for the careful reading and for pointing out several typos and unclear statements. We corrected all typographical issues in the revised manuscript.
>
> ## Minor Issue 5
> > "Form small time steps, the samples from each class are embedded in well separated areas, while they gradually become more random as time increases." This is unclear. Is t representing size of time steps or time? (Or are they confounded.)
>
> Regarding the terminology of “small time steps”, we clarify that this always refers to the value of the continuous time variable t, not to the discretization step size. As described in the manuscript, CoVAE does not have an explicit forward kernel; instead, the notion of time is implicitly induced by the weighting functions $\beta(t)$ and $\lambda(t)$. Empirically (see Figure 4), the effective SNR of the latent space decreases monotonically with t. Thus, each time value represents a position along an implicitly defined latent forward process.
>
> ## Minor Issue 3
>
> > Figure 2 is confusing. I suggest you explain what the objects in the future are one by one, starting from the left. E.g. is "In Diffusion and Consistency" about the first picture or the first two? Epsilon_psi is in in the figure but is in the caption. "in this case we use a dashed line" This is confusing.
>
> Regarding Figure 2, we thank the reviewer for highlighting potential sources of confusion. To summarize the intended interpretation:
> - The left panel (DSM) depicts classical diffusion: the red path is the forward process, and the model learns the average denoiser (blue/purple arrows).
> - The center panel (CM) depicts consistency models: the same red forward path is used, but the model leverages bootstrapping by matching the prediction at time t to the prediction at the previous time step.
> - The right panel (CoVAE) mirrors the CM structure, but with two key differences:
>  (i) the dashed yellow line indicates that the forward process is implicitly defined by the time-conditioned encoder rather than by a closed-form diffusion kernel;
>  (ii) the latent variables live in compressed space, and the consistency model acts as the decoder.
> Apart from this change in the nature of the forward trajectory, the reconstruction objective is structurally the same as the discrete CM bootstrapped loss.
>
> We hope this explanation clarifies the intended meaning of the figure, and we welcome further suggestions for improving its readability.
>
> ## Weakness 2
> > the models compared with are valuable but not state-of-the-art in terms of performance
>
> Regarding the concern that CoVAE does not yet match state-of-the-art diffusion or adversarial models: this is an important point, and we agree. However, we note that the initial versions of diffusion and consistency models also started far below SOTA GANs, and only became competitive after significant refinements and scaling.
>
> Similarly, CoVAE currently lags behind diffusion-based SOTA models in absolute sample fidelity, but achieves state-of-the-art performance among single-stage autoencoder-based generative models, those that do not rely on learned priors or multi-stage latent diffusion. Moreover, because CoVAE is fundamentally an autoencoder, it should not be evaluated solely on raw sample fidelity, but also on its ability to learn a structured and controllable latent space, where it performs comparably to existing autoencoder-based methods. Our aim is to demonstrate that generative autoencoders, trained end-to-end, without complex priors, can be revitalized through consistency-style training. We believe CoVAE opens a promising direction toward scalable and efficient generative modeling in latent space.
>
> A more extensive comparison to modern approaches is provided in Appendix A and Table 5.

---

> > ### Comment · Reviewer_ovEJ · 2025-11-26
> >
> > Thanks to the authors for their response. I have read the response to my review, as well as the other reviews and responses. My concern is whether there is a scaling path, and whether the paper makes a clear case. I am not convinced by the evidence in the paper.

---

### Official Review · Reviewer_b5HX · 2025-10-31

**Soundness:** 3
**Presentation:** 3
**Contribution:** 3
**Rating:** 4
**Confidence:** 4

**Summary:**

The paper proposes CoVAE, training VAEs using consistency loss, in similar fashion as the consistency models. The encoder is parametrized with the time step in addition to the input, and the decoder is trained to map the latent representations from any time step to the original image. The proposed method shows improved performance among the state-of-the-art VAE methods, but still lags behind diffusion models.

**Strengths:**

- The proposed idea in CoVAE to use consistency training in VAEs is novel and interesting.
- The performance among VAEs is better, and CoVAE also offers the option to trade off efficiency and performance with multi-step generation.

**Weaknesses:**

- There is limited insight into the fundamental difference consistency training brings in VAEs that leads to performance improvements. While iterative denoising is intuitively justified in diffusion-based or consistency models—where the coupling between latent variables and data points is unknown, it is less clear in the case of VAEs, where the latent variable corresponding to a given data point can be obtained through the encoder.
- In Section 2.1, the authors mention the prior hole problem, which is a fundamental problem with VAEs when used for generation. It is not discussed whether CoVAE mitigates the problem or if CoVAE has any impact on it.
- At lines 372/373 it is mentioned that a patch-based adversarial loss is used, and it should be clear in the performance tables about the role of this additional loss. While Table 1 has  CoVAE with and without patch-based losses, Table 2 does not include it.
- I would recommend using higher resolution images and the Imagenet dataset for experiments, as it is a fundamental benchmark for image generation.

I believe the paper would benefit from deeper insights into why consistency training is necessary and what conceptual effect it introduces. Additionally, the experimental section could be strengthened through refinements involving higher-resolution datasets, such as ImageNet. I would be open to reconsidering my evaluation if these aspects are addressed.

**Questions:**

-

---

> ### Author Response · Authors · 2025-11-21
>
> We appreciate the reviewer’s request for deeper clarification, as this is indeed the core conceptual contribution of our work.
> ## Weaknesses 1 and 2
> > - There is limited insight into the fundamental difference consistency training brings in VAEs that leads to performance improvements. While iterative denoising is intuitively justified in diffusion-based or consistency models—where the coupling between latent variables and data points is unknown, it is less clear in the case of VAEs, where the latent variable corresponding to a given data point can be obtained through the encoder.
> > - In Section 2.1, the authors mention the prior hole problem, which is a fundamental problem with VAEs when used for generation. It is not discussed whether CoVAE mitigates the problem or if CoVAE has any impact on it.
>
> The key insight is that latent variables learned by a β-VAE behave analogously to latent states produced by the forward process of diffusion/consistency models. While it is true that, in a VAE, the coupling between data and noise is explicitly parametrized by the encoder, the sampling in the reparameterization trick plays the same qualitative role as the diffusion forward kernel.
>
> For small β, the encoder produces almost deterministic latents. The noise injected during the reparametrization trick minimally perturbs the mean, so these latents are easy to decode. However, this regime is precisely where the prior hole problem emerges: the aggregate posterior fails to match the prior, and generative performance is weak despite good reconstruction.
>
> For large β, the opposite occurs: the reparametrization trick removes most information about the encoded mean, yielding highly ambiguous posterior samples. In this case, the prior is well covered, but reconstruction quality collapses, and blurriness arises because many inputs map to overlapping, noisy latent codes. This mirrors the behavior in score-based diffusion models at big t, where x-predictions regress toward the dataset mean.
>
> Thus, a time-dependent β-VAE, as in Eq. 10, implicitly traces a trajectory analogous to the diffusion forward process: low-noise latents near data for small β, and increasingly noisy latents approaching the prior as β grows.
>
> The difficulty is that, unlike diffusion models, a VAE cannot leverage these predictions through an iterative reverse process. However, consistency models solve this exact problem: they bootstrap predictions from early time steps toward later ones to construct a one-step mapping that bypasses iterative denoising. Our method applies the same principle, but in latent space. By replacing the VAE reconstruction loss with a latent consistency loss, we propagate the clean reconstruction signal from small β to large β, enabling a well-covered latent prior without sacrificing reconstruction fidelity.
>
> This directly resolves the prior-hole problem: for large β, the latent distribution matches the prior, but reconstructions are no longer blurry because the decoder objective has been bootstrapped using the consistency objective. Our empirical findings, particularly Figures 3, 4, and 7, strongly support this interpretation.
>
> We hope this explanation clarifies the conceptual motivation behind CoVAE. We would be happy to further elaborate if helpful.
>
> ## Weakness 3
> > At lines 372/373 it is mentioned that a patch-based adversarial loss is used, and it should be clear in the performance tables about the role of this additional loss. While Table 1 has CoVAE with and without patch-based losses, Table 2 does not include it.
>
> We added the results of CoVAE on CelebA without adversarial loss in Table 2 (right).

---

> ### Comment · Reviewer_b5HX · 2025-11-22
> **Response to Rebuttal**
>
> I would like to thank the authors for the detailed explanation. The explanation addresses my initial concern and provides deeper insight into the motivation behind CoVAE. Including these details would strengthen both the clarity and motivation in the paper.
>
> I believe the manuscript could better highlight this perspective and more clearly articulate the authors’ contribution. In the abstract, the authors note that state-of-the-art generative models typically rely on a two-stage VAE–Diffusion training procedure, which is computationally intensive, and that CoVAE challenges this paradigm. Given this framing, it is natural for readers to expect a detailed comparison of CoVAE with these two-stage methods in terms of both performance and efficiency, as well as other relevant aspects.
>
> I suggest emphasizing more clearly the specific advantages of CoVAE compared to SOTA diffusion models. While the experiments in the appendix show that CoVAE may not outperform diffusion models in raw generative quality, it offers a significant advantage: it can simultaneously mitigate the prior-hole problem while maintaining strong generative performance. In contrast, the VAE component in two-stage methods suffers from prior-hole issues. The authors should highlight better the implications of this advantage: for example, maybe it enables more effective manipulation of latent embeddings - learning disentangled features or modifying latent codes for conditional generation, while still producing high-quality images. Such latent-space control may be less reliable in models where the prior-hole problem is prominent. Highlighting such potential implications including relevant experiments for support would better convey the unique contribution and practical utility of CoVAE.

---

> > ### Author Response · Authors · 2025-11-25
> >
> > We thank the reviewer for the constructive feedback and for the helpful suggestions on how to further improve the clarity of our contribution.
> >
> > While two-stage VAE-Diffusion pipelines are indeed the main modern use of VAEs, the training of the autoencoder is conceptually separate from that of the diffusion model. For this reason, we believe the cleanest and most direct comparison is between CoVAE and its VAE counterpart as a generative autoencoder, rather than with full two-stage systems whose performance is dominated by the diffusion model.
> > That said, we agree that CoVAE offers an important advantage relative to the VAE component used in two-stage diffusion pipelines: it mitigates the prior-hole problem while still producing high-quality samples. This yields a more reliable and structured latent space, which can in principle support more effective latent manipulation, feature editing, or conditional generation than standard VAEs used in latent diffusion. We will revise the manuscript upon acceptance to better highlight this perspective and clarify the practical implications of a more coherent latent space. Investigating these capabilities empirically is a promising direction, and we plan to explore such latent-space experiments in future work.

---

### Official Review · Reviewer_Ptc5 · 2025-10-31

**Soundness:** 3
**Presentation:** 3
**Contribution:** 2
**Rating:** 4
**Confidence:** 4

**Summary:**

The paper proposes CoVAE, a single-stage generative autoencoding framework that unifies a time-dependent \beta-VAE with consistency training. The encoder produces progressively noised latents via a time-dependent KL weight \beta(t); the decoder is trained with a latent consistency loss (bootstrapping adjacent times) with a denoiser-style term. This enables one- or few-step generation without a learned prior. On MNIST, CIFAR-10, and CelebA-64, CoVAE improves FID over standard VAE and \beta-VAE baselines and outperforms strong single-stage VAEs (NVAE, DC-VAE); adding a lightweight adversarial term further improves FID and reconstruction.

**Strengths:**

1. The paper formulates the VAE reparametrization as a time-indexed “forward process” in latent space and replaces standard reconstruction with a discrete consistency objective that bootstraps from early times. The method section and algorithmic details substantiate this bridge.

2. CoVAE generates in one step and can optionally do few-step refinement by re-encoding/re-denoising at intermediate t. This is a practical departure from the common “VAE + latent diffusion/flow” recipe.

3. The paper shows promising reconstruction and generation capacity of CoVAE in experiments. On CIFAR-10, CoVAE (1-step) improves FID over NVAE and DC-VAE. On MNIST, CoVAE simultaneously improves generation and reconstruction over β-VAE.

**Weaknesses:**

1. A major concern is the applicability of the proposed approach, both to future research and real-world application. While CoVAE aims to unify VAE and the diffusion process for generation tasks in one single stage, it neglects text (or class) conditioning in modeling and implementation for image generation, which is crucial in current generative models. The paper compares CoVAE with standard VAE and demonstrates its advantages. However, standard VAE can be readily used to modeling visual signals in the latent space, for further diffusion-based generation. It is not clear how the time-dependent latents in CoVAE can be refined or utilized in downstream tasks.

2. Experiments show results on relatively low-resolution (up to 64x64) image generation tasks. More convincing results are missing to show how CoVAE performs on high-resolution tasks and how it compares with strong baselines such as GANs.

**Questions:**

1. Are there principled ways to design \beta(t) and \lambda(t)? How sensitive is empirical performance to their values?

2. What is the compute and inference time of CoVAE? And what about the training efficiency and scalability of CoVAE?

3. Would the generation quality consistently improve if the sampling steps are increased?

---

> ### Author Response · Authors · 2025-11-21
>
> We thank the reviewer for the thoughtful comments. Below, we address the identified weaknesses and questions.
>
> ## Weakness 1
> >A major concern is the applicability of the proposed approach, both to future research and real-world application. While CoVAE aims to unify VAE and the diffusion process for generation tasks in one single stage, it neglects text (or class) conditioning in modeling and implementation for image generation, which is crucial in current generative models. The paper compares CoVAE with standard VAE and demonstrates its advantages. However, standard VAE can be readily used to modeling visual signals in the latent space, for further diffusion-based generation. It is not clear how the time-dependent latents in CoVAE can be refined or utilized in downstream tasks.
>
> Class conditioning can be incorporated into CoVAE in a straightforward manner, either by conditioning only the decoder, or both the encoder and decoder. We did not include class-conditional experiments in our benchmarks primarily because class conditioning is not commonly reported on the datasets we evaluate (MNIST, CIFAR-10, CelebA 64). Nonetheless, we expect the extension to be straightforward in practice.
>
> A more nuanced case concerns Classifier-Free Guidance (CFG). While CFG-style conditioning can be implemented for Consistency Models, it remains an active research area due to the subtleties of integrating null-conditioning into one-step or few-step generative architectures. Although highly relevant for practical applications, we view CFG as orthogonal to the core contribution of CoVAE, and we believe it merits dedicated investigation beyond the scope of this work.
>
> Finally, regarding downstream use of latents: CoVAE produces a structured latent representation similar to standard VAEs, but how these time-dependent latents might best be leveraged for downstream tasks (e.g., classification, retrieval, or latent generation) is an interesting question that we leave for future work.
>
> ## Question 1
> > Are there principled ways to design \beta(t) and \lambda(t)? How sensitive is empirical performance to their values?
>
> In standard diffusion and consistency setups, weighting functions are often derived from the signal-to-noise ratio (SNR) of the forward process. In CoVAE, the latent forward process is learned, and its SNR is therefore not known a priori. For this reason, our choices of β(t) and λ(t) were empirically tuned. While we could achieve comparable performance with other choices, we found this configuration to work best. A promising future direction is to measure the latent SNR during training and use it to adaptively adjust β(t) and λ(t), potentially yielding more principled schedules. Another direction is the simplified formulation discussed in Section E.1: while it performed worse in our experiments, it substantially simplifies the overall algorithm (at the cost of removing the learned data-noise coupling) and may serve as a basis for future improvements.
>
> ## Question 2
> > What is the compute and inference time of CoVAE? And what about the training efficiency and scalability of CoVAE?
>
> Compute and inference time depend on hardware, but Table 3 reports the decoder and encoder GFLOPs for our configurations. Scaling CoVAE further is largely tied to architectural design choices; in this work, we adopted the autoencoder backbone from Stable Diffusion without additional architectural optimization. Exploring architectures specifically tailored for CoVAE is an important future direction for scaling.
>
> ## Question 3
> > Would the generation quality consistently improve if the sampling steps are increased?
>
> We report multi-step results in Table 4. While performance improves when increasing the number of sampling steps, we observe diminishing returns after a small number of iterations. This behavior mirrors known limitations of multi-step sampling in Consistency Models, where most performance gains occur in the first few steps.

---

### Author Response · Authors · 2025-11-21
**General Answer**

We sincerely thank all reviewers for their thoughtful and constructive feedback.

We are encouraged to see that all reviewers recognized the novelty of CoVAE, specifically the integration of time-dependent latent VAEs with consistency-style training, and acknowledged the empirical benefits over both standard VAEs and strong single-stage VAE baselines. We also appreciate that several reviewers highlighted the practicality of one and few-step sampling and the conceptual contribution of unifying autoencoding and consistency training within a single framework.

A recurring request concerns evaluating CoVAE on higher-resolution datasets such as ImageNet. We agree that large-scale experiments are an important and informative test of scalability. However, such experiments are beyond our current computational budget. Importantly, our goal in this paper is to establish the viability of a single-stage generative autoencoding framework trained end-to-end, without complex priors or diffusion-style sampling. For this purpose, we selected datasets that allow extensive ablations, controlled comparisons, and fast iteration within our resource constraints. Across these benchmarks, CoVAE consistently delivers strong empirical results, substantially outperforming equivalent VAE and β-VAE architectures and improving over strong one-stage VAE baselines (NVAE, DC-VAE), even without employing hierarchical priors or complex flows.

---

### Meta-Review · Area_Chair_FFc9 · 2026-01-13

**Summary:**

There was general agreement among reviewers that the idea is interesting and nontrivial: unifying \beta-VAEs and consistency models into a single-stage generative autoencoder with fast (one/few-step) sampling. Formulation is sound, experiments careful, and empirical gains over standard and strong single-stage VAEs consistent.

However, there were several concerns that make acceptance hard to justify. The work doesn't convincingly demonstrate a scaling path: experiments are restricted to low-resolution datasets (<=64x64), and there is no evidence that the method can compete with or meaningfully complement modern large-scale diffusion or latent diffusion systems. Reviewers also felt that comparisons stop short of SOTA generative models, making it difficult to assess the broader impact. There were also concerns regarding reliance on empirically tuned schedules, lack of likelihood-based evaluation, and limited empirical validation of claimed benefits such as improved latent-space structure or downstream utility.

**Reviewer Concerns:**

Though the rebuttal addressed some of the clarity concerns, e.g., Reviewer b5HX’s questions about consistency training and prior-hole problem, the core outstanding issues remained. In particular, the lack of high-res or large-scale experiments, unclear scalability, limited comparison to modern diffusion or latent-SDE-based approaches, and absence of strong empirical evidence that CoVAE's claimed advantages translate into concrete downstream or practical gains. Reviewer ovEJ explicitly remained unconvinced after the rebuttal, citing insufficient evidence on scaling path.

**Reviewer Scores:**

Reviewer Ptc5 score (4) would have likely remained unchanged. The rebuttal addressed technical questions but didn't resolve concerns about applicability, conditioning, and large-scale relevance. Reviewer b5HX score (4) may have raised to weak accept or high borderline as they explicitly stated that their main conceptual concern was resolved, though they still desired stronger positioning against diffusion models. Reviewer ovEJ score (4) would have likely remained unchanged. The reviewer reiterated their skepticism about scaling and impact after reading the rebuttal. Reviewer DFFZ score (2) was unlikely to change due to fundamental concerns about evaluation, comparisons, and scalability not being resolved.

---

### Decision · Program_Chairs · 2026-01-26

Reject